# One Anomaly to Catch Them All: Graph-Based One-Shot Anomaly Detection via Grad-Conf

## Abstract

Graph anomaly detection faces challenges of scarce labeled samples and concealed anomalous features. Although recent graph-based models have shown potential, their reliance on extensive supervisory signals limits their effectiveness in real-world scenarios. To address this issue, we propose the GradConf framework, which enables robust anomaly detection under extremely low supervision. This framework constructs a graph structure where nodes represent entities and edges denote associative relationships, enhancing model robustness through view enhancement and consistency learning. Based on this, our key contributions are as follows: (1) Proposing a Gradient-Confidence Aware Loss that dynamically balances positive and negative samples by combining global training gradients with instance-level confidence; (2) Designing a Pseudo-label Clustering Self-Correction module that iteratively optimizes pseudo-label quality via learnable clustering centers and a structure-aware self-correction mechanism; (3) Introducing a Logits Adversarial Perturbation strategy that injects perturbations in the logit space to improve the model's sensitivity to anomalies and generalization ability under low supervision. Experiments on six real-world datasets demonstrate that GradConf, using only a single pair of labeled samples, can achieve or even outperform fully supervised methods, verifying its effectiveness and practicality.

## 1 Introduction

In recent years, with the rapid development of the Internet, graph-structured data has become ubiquitous. Graph Anomaly Detection (GAD) aims to identify a small number of "atypical graph objects" in graph-structured data that differ significantly from the majority of objects (Qiao et al., 2025; Han et al., 2022; Akoglu et al., 2015). It is widely applied in various real-world scenarios, such as detecting money laundering activities in financial networks (Huang et al., 2022), identifying malicious reviews in review networks (Li et al., 2019), and uncovering bot accounts on social platforms (Guo et al., 2021). However, due to the complex structure and attributes of graph data (Liu et al., 2023; Ma et al., 2023; Qiao & Pang, 2023), identifying anomalous nodes in graphs poses significant challenges. Although GAD is crucial for maintaining the integrity of these systems, effectively addressing this problem still faces numerous challenges (Lin et al., 2024), including label imbalance (Liu et al., 2022a; Du et al., 2024; Gupta et al., 2023), relational camouflage (Platonov et al., 2023; Huang et al., 2025; Liu et al., 2020), and feature heterophily (Gao et al., 2023a; Dong et al., 2025b).

Based on the availability of supervision information, GAD mainly encompasses three mainstream paradigms: unsupervised, semi-supervised, and fully supervised learning. Unsupervised methods excavate inherent data patterns through strategies such as data reconstruction (Ding et al., 2019; Fan et al., 2020), self-supervised learning (Chen et al., 2020; Liu et al., 2021b; Meng et al., 2023), and one-class homophily modeling (Qiao & Pang, 2023). While they are suitable for extreme label-free scenarios, they fail to utilize easily accessible node labels to calibrate representations. Moreover, the lack of real anomaly domain information may lead to learning biases, ultimately resulting in sub-optimal performance. Semi-supervised and fully supervised methods, although leveraging labels to improve performance, generally require the simultaneous acquisition of a large number of labeled normal and anomalous nodes (Dou et al., 2020; Liu et al., 2021a; Peng et al., 2021; Wu et al., 2024; Tang et al., 2023; Wang et al., 2025). However, the rarity of anomalous nodes and the high cost of

their annotation trap these methods in the dilemma of "theoretically effective but practically inapplicable". Additionally, the strong dependence of current methods on the types of annotated anomalous samples makes it more difficult for them to address the demand for heterogeneous anomaly detection. Therefore, we focus on a more realistic yet challenging scenario: when there is only one labeled anomalous node, can the model still capture its key anomaly-indicating features and generalize to identify unknown anomalies with sparse distributions and subtle patterns?

Specifically, we propose GradConf, a graph anomaly detection framework tailored for extremely low-supervision environments. Centered on the nodes in the graph, this framework constructs a general graph structure using the associative relationships between nodes as edges. To mitigate optimization bias caused by extreme class imbalance, we introduce a Gradient-Confidence Aware Loss (GCAL), which adaptively balances the contributions of positive and negative sample pairs by coupling global gradient signals with instance-level confidence. To address the instability of pseudo-labels under limited supervision, we propose a Pseudo-label Clustering Self-Correction (PCSC) module. It continuously refines label distributions via learnable cluster centers and integrates a structure-aware self-revision mechanism to suppress error propagation induced by clustering noise, thus improving pseudo-label reliability and generalization to unlabeled samples. Furthermore, considering the rarity and concealment of anomaly patterns, we incorporate a Logits Adversarial Perturbation (LAP) mechanism within the encoder to enhance the discriminability of learned features and improve the model's sensitivity to anomalous instances.

Overall, the main contributions of this paper are as follows:

(1) To the best of our knowledge, this work is the first to systematically focus on graph anomaly detection under the practical constraints of extremely low labeling rates, weak graph homophily, and extreme class imbalance, bridging the gap between existing research assumptions and real-world deployment scenarios;

(2) We propose GradConf, a plug-and-play contrastive training framework that integrates gradient-aware loss, clustering-based pseudo-label correction, and adversarial perturbation to tackle class imbalance, pseudo-label noise, and anomaly concealment under minimal supervision;

(3) Systematic experiments are conducted on graph datasets from six real-world scenarios. The results demonstrate that GradConf, using only one pair of labeled samples, can achieve performance comparable to or even surpassing that of fully supervised methods, confirming its effectiveness and feasibility under extreme conditions.

## 2 RELATED WORK

**Graph Anomaly Detection.** Graph Anomaly Detection (GAD) aims to identify nodes, edges, or subgraphs in a graph that deviate from normal patterns, and it serves as a core technology for risk prevention and control in complex systems (Ma et al., 2021). Early shallow methods (Li et al., 2017; Peng et al., 2018; Perozzi & Akoglu, 2016) are limited by their representational capacity, making it difficult to capture the complex semantics and high-order correlations of graphs. With the development of GNNs, deep GAD methods have become the mainstream. Reconstruction-based methods from the spatial perspective (Ding et al., 2019; Fan et al., 2020), correlation mining methods (Ma et al., 2023), and signal analysis methods from the spectral perspective (Liu et al., 2021b; Tang et al., 2022; Gao et al., 2023a) have all improved performance through the adaptive aggregation of topological and attribute information. However, recent specialized models (Gao et al., 2023b; Gong et al., 2023; Wang et al., 2023b; Tang et al., 2023) still fail to overcome two core bottlenecks. First, they generally rely on sufficient supervision signals (Liu et al., 2022b; Tang et al., 2023), which conflicts with the requirement of "scarce labeled samples" in real-world scenarios. Second, their adaptability to scenarios with "concealed anomaly features" is insufficient. Generative methods (Ding et al., 2021; Chen et al., 2020) fail to fully utilize topological information, leading to a significant mismatch between the distribution of pseudo-anomalies and real anomalies (Zenati et al., 2018; Ngo et al., 2019). Furthermore, most existing methods focus on single-level anomaly detection (Liu et al., 2023; Dong et al., 2023), fail to capture inter-level collaborative correlations and thus miss cross-level hidden anomalies. This highlights the necessity of the GradConf.

**Semi-supervised Learning on Graphs.** Semi-supervised learning on graphs addresses node classification with limited labeled nodes by leveraging topology and unlabeled information. Early works

commonly adopt message passing frameworks to propagate label signals across neighborhoods (Kipf & Welling, 2017; Yang et al., 2016; Zhu et al., 2003). To enhance representation learning, recent studies explore adversarial training (Dai et al., 2018; Jin et al., 2021; Xu et al., 2022), data augmentation (You et al., 2020; Wang et al., 2020; Sui et al., 2023), pseudo-labeling (Wang et al., 2023a), virtual connections (Xie et al., 2023), and information regularization (Zhang et al., 2025; Chen et al., 2021) to reduce oversmoothing and distribution bias. Among them, contrastive learning has gained increasing attention for its ability to extract structure-aware representations without heavy reliance on labels (Yin et al., 2023; Veličković et al.; Liu et al., 2022c; Li et al., 2021; Bo et al., 2023), often through augmentation-consistent training or feature-level alignment. Recent efforts also explore curriculum-aware sampling (Zhang et al., 2023), consistency-enforcing teacher-student frameworks (Chang et al., 2023; Liu & Zhang, 2021), and prompt-based or meta-knowledge-driven tuning strategies for efficient graph adaptation in low-label regimes (Holtz et al., 2024; Shao et al., 2024). Despite these advances, many approaches rely on implicit assumptions such as label proximity or structural regularity, which are often violated in real-world graph anomaly detection scenarios characterized by sparse supervision and irregular connectivity patterns.

## 3 METHODOLOGY

### 3.1 PROBLEM SETUP

**One-shot Graph Anomaly Detection:** Graph anomaly detection is a binary classification task on graph-structured data $D = \{(t_1, y_1), \ldots, (t_N, y_N)\}$, where $t_i$ is a node with feature vector $X_i \in \mathbb{R}^d$, and $y_i \in \{0, 1\}$ is its label (1 for anomaly, 0 for normal). GNNs are commonly used by constructing a graph $G(V, E, X, A)$, where $V = \{v_1, \ldots, v_N\}$ are nodes, $E \subseteq V \times V$ are edges, $X \in \mathbb{R}^{N \times d}$ is the node feature matrix, and $A \in \{0, 1\}^{N \times N}$ is the adjacency matrix. We define this as: a binary semi-supervised node classification problem on a graph $G = (V, E, X, A)$. The labeled set $\mathcal{D}_L = \{(x_f, 1), (x_l, 0)\}$ contains one anomalous and one normal node. A large unlabeled set $\mathcal{D}_U = \{x_{u_1}, \ldots, x_{u_M}\}$ (where $M \gg |\mathcal{D}_L|$) is also utilized. Let $\mathcal{V}_l$ and $\mathcal{V}_u$ be node sets for $\mathcal{D}_L$ and $\mathcal{D}_U$. The goal is to learn $f : V \to \{0, 1\}$ using $G, \mathcal{D}_L, \mathcal{D}_U$ to predict labels for an unseen test set $Q = \{x_{q_1}, \ldots, x_{q_P}\}$ (nodes $\mathcal{V}_q$, disjoint from $\mathcal{V}_l, \mathcal{V}_u$).

### 3.2 OVERVIEW

GradConf first generates two augmented graph views(Zhao et al., 2021), $G'_k = t_k(G)$, from the original graph $G = (V, E, X, A)$ using independent augmentation operators $t_1(\cdot)$ and $t_2(\cdot)$. A shared-parameter GNN encoder $f_\theta$ processes these views for representative node embeddings $H'_k$, guiding initial training with supervised signals from sparse labeled data $\mathcal{D}_L$. This phase includes three key losses: a consistency loss $\mathcal{L}_{cons}$ ensuring consistent node embeddings across augmented views; a supervised negative log likelihood loss(Yao et al., 2020) $\mathcal{L}_{sup}$, and a supervised contrastive loss $\mathcal{L}_{cls}$ on labeled nodes. These three losses collectively form the base loss $\mathcal{L}_{base}$ (Figure1, left), detailed in the APPENDIX A.7. Building on this foundation, GradConf introduces GCAL, PCSC, and LAP, as illustrated on the right side of Figure 1.

### 3.3 GRADIENT-CONFIDENCE AWARE LOSS

Effective pseudo-label is often challenged by noise and class imbalance(Zou & Cheng, 2024; Xiang et al., 2023; Tang et al., 2022; Xiang et al., 2025; Dou et al., 2020; Shi et al., 2022; Liu et al., 2021a). While existing methods(Qian et al., 2022; Xiao et al., 2023; Qi et al., 2024; Miao et al., 2024; Yang & Xu, 2020) employ re-weighting or resampling, they often lack full adaptivity to both class-level imbalance and sample-specific difficulty as revealed by gradients. To address this, we introduce $\mathcal{L}_{gcal}$ for robust pseudo-label learning.

GCAL first compute the predicted probability $p_{t,i}$ of sample $i$ for its pseudo-label $\hat{y}_i^{pl}$ at view k. Its standard cross-entropy loss $CE_i = -\log(p_{t,i})$ is then modulated by three weighting components: First, Focal Weighting $w_{focal,i} = (1 - p_{t,i})^{\gamma_f}$ (where $\gamma_f$ is set to 4/5), emphasizes hard-to-classify samples based on prediction confidence. Second, Dynamic Class Balancing Weight $w_{class,i}$ estimates and adjusts class weights based on the K% (set to 10%) hardest samples (highest $CE_i$). Let

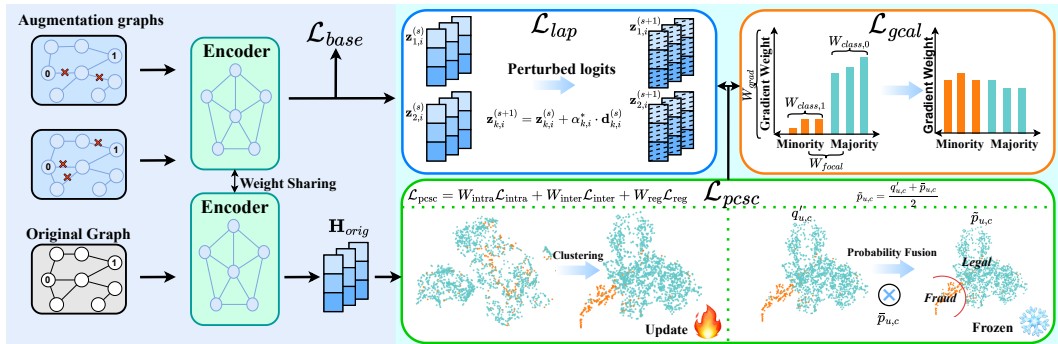

Figure 1: Global overview of GradConf. The Gradient-Confidence Aware Loss (GCAL) uses gradient-confidence to balance imbalanced samples. Pseudo-label Clustering Self-Correction (PCSC) refines pseudo-labels via clustering and self-revision for few-shot learning. Logits Adversarial Perturbation (LAP) perturbs logits adversarially to boost anomaly sensitivity and generalization.

$n_c^{hard,(t)}$ be the count of class c samples among the hardest examples at training step t. The smoothed effective count for class c, $N_c^{(t)}$, is updated using a momentum term $\beta$ (set to $9/10$).

$$N_c^{(t)} = \beta N_c^{(t-1)} + (1 - \beta) n_c^{hard,(t)}. \tag{1}$$

The weight $\alpha_c$ for class $c$ is then calculated as:

$$\alpha_c = \left(\frac{1}{N_c^{(t)}}\right)^{\gamma_{cb}}, \tag{2}$$

where $\gamma_{cb}$ (set to $1/2$) governs the intensity of class balancing. Class weights $\alpha_c$ are normalized ($\sum_c \alpha_c = C$, where $C$ is the number of classes). For a sample $i$ with pseudo-label $\hat{y}_i^{pl}$, its class balancing weight is $w_{class,i} = \alpha_{\hat{y}_i^{pl}}$. Third, as samples with larger gradient magnitudes often reside near the decision boundary or represent instances where the model is uncertain, we compute the L2 norm of gradient magnitude weight $w_{grad,i}$ with respect to the input logits $\mathbf{z}_{k,i}$:

$$g_{k,i} = \left\|\nabla_{\mathbf{z}_{k,i}} \mathrm{CE}(\mathrm{softmax}(\mathbf{z}_{k,i}), \hat{y}_i^{pl})\right\|_2. \tag{3}$$

Initially, the class balancing weight $w_{class,i}$ and the gradient magnitude weight $w_{grad,i}$ are multiplied to form a class-aware difficulty weight, $w_{diff,i}$ ($w_{\mathrm{class},i} \cdot w_{\mathrm{grad},i}$), then normalized by:

$$\tilde{w}_{\mathrm{diff},i} = \frac{w_{diff,i}}{\frac{1}{|\mathcal{D}_{PL}|} \sum_{j \in \mathcal{D}_{PL}} w_{diff,j}}. \tag{4}$$

Subsequently, this normalized class-aware difficulty weight $\tilde{w}_{\mathrm{diff},i}$ is multiplied by the focal weight $w_{focal,i}$. This step refines the overall weighting by incorporating the sample-level hardness (prediction confidence) as captured by the focal term, yielding a preliminary final weight $w_{\mathrm{final\_raw}}$:

$$w_{final\_raw,i} = \tilde{w}_{diff,i} \cdot w_{focal,i}, \tag{5}$$

and

$$\tilde{w}_{\mathrm{final},i} = \frac{w_{final\_raw,i}}{\frac{1}{|\mathcal{D}_{PL}|} \sum_{j \in \mathcal{D}_{PL}} w_{final\_raw,j}}. \tag{6}$$

The final $\mathcal{L}_{gcal}$ is defined as:

$$\mathcal{L}_{gcal}(\mathbf{Z}_k, \hat{\mathbf{Y}}) = \frac{1}{B} \sum_{i \in \mathcal{D}_{\mathrm{PL}}} \tilde{w}_{\mathrm{final},i} \cdot \mathrm{CE}_i, \tag{7}$$

where $\mathbf{Z}_k = \{\mathbf{z}_{k,i}\}_{i \in \mathcal{D}_{PL}}$ represents the set of logits for samples in $\mathcal{D}_{PL}$ from the $k$-th view, and $\hat{\mathbf{Y}} = \{\hat{y}_i^{pl} \mid i \in \mathcal{D}_{PL}\}$ is the set of their corresponding pseudo-labels.

The total loss from pseudo-labels is averaged across the two augmented views:

$$\mathcal{L}_{\mathrm{gcal}}^{\mathrm{total}} = \frac{1}{2} \left( \mathcal{L}_{\mathrm{gcal}}(\mathbf{Z}_1, \hat{\mathbf{Y}}) + \mathcal{L}_{gcal}(\mathbf{Z}_2, \hat{\mathbf{Y}}) \right). \tag{8}$$

### 3.4 PSEUDO-LABEL CLUSTERING SELF-CORRECTION

To address the underutilization of unlabeled graph data by existing anomaly detectors (Zou & Cheng, 2024)(Tang et al., 2022)(Xiang et al., 2023)(Xiang et al., 2025)(Shi et al., 2022)(Dou et al., 2020)(Liu et al., 2021a) in one-shot scenarios, PCSC utilizes learnable cluster centroids, initialized based on original node embeddings $H_{orig}$ and optimized via $\mathcal{L}_{pcsc}$ aimed at enhancing cluster compactness and separability. These evolving centroids then guide the soft cluster assignments $q_{u,j}$ for unlabeled nodes $u \in \mathcal{D}_U$ computed using Gumbel-Softmax(Jang et al., 2016), considering Euclidean distanced ($\mathbf{h}_{orig,u}, \mathbf{c}_j$) and Gumbel noise(Jang et al., 2016) $g_j$. Then, $q_{u,j}$ are fused with the GNN model's predictions to generate pseudo-labels $\hat{y}^{pl}$ for the GCAL and LAP strategies.

$$q_{u,j} = \frac{\exp((g_j - d(\mathbf{h}_{orig,u}, \mathbf{c}_j))/\tau_{clus})}{\sum_{k=0}^{C_k-1} \exp((g_k - d(\mathbf{h}_{orig,u}, \mathbf{c}_k))/\tau_{clus})}. \tag{9}$$

where $\tau_{clus}$ (set to 4/5) donates the sharpness of the soft assignments. Node $u$'s raw cluster assignment is $k_u = \arg\max_j q_{u,j}$. To align the clustering probability vector $q_{u,j}$ (for raw clusters) with a semantic $[P(normal), P(anomalous)]$ format for fusion, we use prior knowledge (anomaly as minority): the more populous raw cluster in $\mathcal{K} = \{k_u | u \in \mathcal{D}_U\}$ is normal (class0), the less populous is anomalous (class1), which converts $q_{u,j}$ into the semantically ordered $q'_{u,c} = [q'_{u,0(normal)}, q'_{u,1(anomalous)}]$. The model-derived probability for semantic class $c$, $\bar{p}_{u,c}$, is typically obtained by averaging the softmax outputs from two augmented views:

$$\bar{p}_{u,c} = \frac{1}{2}(\text{softmax}(\mathbf{z}_{1,u})_c + \text{softmax}(\mathbf{z}_{2,u})_c). \tag{10}$$

where $c \in \{0(normal), 1(anomaly)\}$, and $\mathbf{z}_{1,u}, \mathbf{z}_{2,u}$ are the logits from the Encoder for node $u$ on the two augmented views, respectively. Specifically, we first compute an averaged probability $\tilde{p}_{u,c}$ by taking the mean of $q'_{u,c}$ and $\bar{p}_{u,c}$:

$$\tilde{p}_{u,c} = \frac{q'_{u,c} + \bar{p}_{u,c}}{2}. \tag{11}$$

The pseudo-label $\hat{y}^{pl}_i$ is then determined by selecting the class with the highest averaged probability:

$$\hat{y}^{pl}_i = \arg\max_c \tilde{p}_{u,c}. \tag{12}$$

These generated pseudo-labels $\hat{y}^{pl}_i$ for all $i \in \mathcal{D}_U$ form the set $\mathcal{D}_{PL}$, which provides supervisory signals for the $\mathcal{L}_{gcal}$. Crucially, the learnable cluster centroids $\mathbf{C}$ (used to derive $q'_{u,c}$) are continually refined by optimizing $\mathcal{L}_{pcsc}$ and $\mathcal{L}_{lap}$.

Embeddings from original and two augmented views $\mathbf{H}_{all}$ are aligned by $\mathcal{L}_{pcsc}$ towards initial soft assignments $\mathbf{Q}_{orig}$ and centroids $\mathbf{c}_j$ computed only from the less noisy original view $\mathbf{H}_{orig}$. First, the Intra-cluster Loss $\mathcal{L}_{intra}$ encourages embeddings to be close to their assigned cluster centroids. This is measured by the $D_{KL}$ divergence (details in the appendix A.11) computed between the softmax-normalized probability distribution of an embedding $\mathbf{h}_u$ drawn from $\mathbf{H}_{all}$, $P(\mathbf{h}_u) = softmax(\mathbf{h}_u)$, and that of a centroid, $P(\mathbf{c}_j) = softmax(\mathbf{c}_j)$.

$$D_{KL}(P \| Q) = \sum_x P(x) \log \frac{P(x)}{Q(x)}, \tag{13}$$

and

$$\mathcal{L}_{intra} = \mathbb{E}_{(\mathbf{h}_u, \mathbf{q}'_u) \sim (\mathbf{H}_{all}, \mathbf{Q}_{orig})}[\sum_{j=0}^{1} q_{uj} D_{KL}(P(\mathbf{h}_u) \| P(\mathbf{c}_j))]. \tag{14}$$

Second, the Inter-cluster Loss $\mathcal{L}_{inter}$ promotes dissimilarity between different centroids. This is achieved using the symmetric KL divergence, defined as:

$$D_{SKL}(P_i \| P_j) = \frac{1}{2}(D_{KL}(P_i \| P_j) + D_{KL}(P_j \| P_i)), \tag{15}$$

where $P_i = P(\mathbf{c}_i)$ and $P_j = P(\mathbf{c}_j)$ are the softmax-normalized distributions of two distinct centroids.

$$\mathcal{L}_{inter} = -\log \sum_{(i,j), i \neq j} \exp\left(-\frac{D_{SKL}(P(\mathbf{c}_i) \| P(\mathbf{c}_j))}{\tau_c}\right), \tag{16}$$

where $\tau_c$ is a temperature hyperparameter (set to 1/2) that controls the sensitivity to the dissimilarity between centroids.

Third, the Centroid Regularization Loss $\mathcal{L}_{reg}$ regularizes the centroids towards the origin using Smooth L1 loss, to prevent them from becoming excessively large and to mitigate potential overfitting.

$$\mathcal{L}_{\text{reg}} = \mathbb{E}_j[\text{SmoothL1}(\mathbf{c}_j, \mathbf{0})]. \tag{17}$$

These components are weighted and summed to form the total clustering loss. The weights are set as: $w_{intra} = 1.0$; $w_{inter} = \min(0.5, 10.0/(N_p + N_n))$, where $N_p$ and $N_n$ are the number of samples assigned to each raw cluster (0 and 1) and $w_{reg} = 0.01 \times (d_f/16.0)$, where $d_f$ is the feature dimensionality. Thus, the overall clustering loss is:

$$\mathcal{L}_{\text{pcsc}} = w_{\text{intra}}\mathcal{L}_{\text{intra}} + w_{\text{inter}}\mathcal{L}_{\text{inter}} + w_{\text{reg}}\mathcal{L}_{\text{reg}}. \tag{18}$$

### 3.5 LOGITS ADVERSARIAL PERTURBATION

To enhance the models sensitivity to anomalous instances and improve generalization under distributional sparsity, we introduce a novel hierarchical adaptation for logits perturbation to generate adversarial logits $z'_{k,i} = z_{k,i} + \eta_{k,i}$ by iteratively perturbing original logits $z_{k,i}$ of pseudo-labeled sample $i$ (view $k$). Efficacy depends on dynamically adjusted single-step strength $\alpha^*_{k,i}$ and total steps $S^*_{k,i}$. These adapt from base settings ($\alpha_0 = 0.03$, $S_{0\_iter} = 20$) considering class imbalance and sample hardness.

We first calculate preliminary class-aware steps $S'_c$. Minimum $S_{min} = \max(1, int(S_{0\_iter} \cdot 0.5))$ and maximum $S_{max} = int(S_{0\_iter} \cdot 2.0)$ steps are defined. Using observed class counts $N_c$, relative class frequencies $\rho_c = N_c / \sum_j N_j$ and an inverse frequency factor $\phi_{freq,c} = \sqrt{1.0/\rho_c}$ are derived to determine $S'_c$:

$$S'_c = S_{min} + \text{int}\left((S_{max} - S_{min}) \cdot \frac{\phi_{freq,c}}{\phi_{freq,c} + 1.0}\right). \tag{19}$$

For a given sample $i$ with pseudo-label $\hat{y}^{pl}_i$, its preliminary perturbation steps, denoted $S'_{k,i}$ (though independent of view $k$ at this stage), are then set to $S'_{\hat{y}^{pl}_i}$.

First, if class $c$ is the minority class, then a class imbalance-aware scaling factor $\omega_{imb,c} = \min(C_{cap}, (K_{imb})^{0.5})$, where the ratio of majority to minority class frequencies $K_{imb}$ is clamped to the range $[1.0, 10.0]$, and the scaling cap $C_{cap}$ is set to 5.0, thus assigning greater weight to minority classes. Second, a class-average gradient-aware scaling factor $\omega_{grad,c}$ is calculated. Based on the momentum-updated average cross-entropy loss $\bar{\mathcal{L}}_{CE,c}$ for samples in each class $C$, which is then normalized using a softmax function to get a normalized average gradient indicator $\tilde{g}_c$. Then, $\omega_{grad,c} = (1.0 + \tilde{g}_c \cdot \lambda_{cg})$, where $\lambda_{cg}$ (set to 2) controls the weight of class gradient. Finally, the preliminary class-aware strength for class $c$, $\alpha'_c$, is:

$$\alpha'_c = \alpha_0 \cdot \omega_{imb,c} \cdot \omega_{grad,c}. \tag{20}$$

For sample $i$, class-aware strength $\alpha'_{k,i} = \alpha'_{y_i}$. Individual sample difficulty $g_{ind,i}$ is derived from the L2 norm of the gradient of $\mathcal{L}_{CE}$ which is computed using logits $z_{k,j}$ and pseudo-label $y_i$ with respect to $z_{k,i}$ to yield final $S^*_{k,j}$ and $\alpha^*_{k,j}$:

$$g_{ind,i} = \left\|\nabla_{\mathbf{z}_{k,i}}\mathcal{L}_{CE}(\text{softmax}(\mathbf{z}_{k,i}), \hat{y}_i)\right\|_2. \tag{21}$$

Then, the gradient norms are normalized through a Softmax function to obtain normalized individual sample difficulties:

$$\delta_i = \frac{\exp(g_{ind,i})}{\sum_j \exp(g_{ind,j})}. \tag{22}$$

The final adaptive steps and strength are:

$$S^*_{k,i} = int(S'_{k,i} \cdot \delta_i), \tag{23}$$

and

$$\alpha^*_{k,i} = \alpha'_{k,i} \cdot \delta_i. \tag{24}$$

First, calculate the average softmax probability vector $\bar{\pi}_c^{(s)}$ for each class $c$ at iteration step $s$. Then, construct a base direction matrix $\mathbf{V}_{dir}$, where row c (the base direction vector for class $c$, $\mathbf{V}_{dir,c} =$normalize$(\bar{\pi}_c^{(s)}\mathbf{e}_c)$, with $e_c$ being the one-hot encoding vector for class $c$. Let $\pi_{c,i}^{(s)}$ be the $i$-th element of $\bar{\pi}_c^{(s)}$ (the average self-prediction probability for class $c$). A threshold $\tau_{avg\_prob}^{(s)}$ is calculated as the mean of these self-prediction probabilities across all classes. Then, a sign modulation factor $\sigma_c = $sign$(\tau_{avg\_prob}^{(s)} - \bar{\pi}_{c,c}^{(s)})$. For a sample $i$ with pseudo-label $\hat{y}_i^{pl}$ its perturbation direction $\mathbf{d}_{k,i}^{(s)}$ at iteration $s$ and view k is:

$$\mathbf{d}_{k,i}^{(s)} = \mathbf{v}_{dir,\hat{y}_i^{pl}} \cdot \sigma_{\hat{y}_i^{pl}}. \tag{25}$$

Subsequently, an iterative process is employed to apply the perturbation:

$$\mathbf{z}_{k,i}^{(s+1)} = \mathbf{z}_{k,i}^{(s)} + \alpha_{k,i}^* \cdot \mathbf{d}_{k,i}^{(s)}, \quad \mathbf{s} = 0, \ldots, \mathbf{S}_{k,i}^* - 1 \tag{26}$$

where $\mathbf{z}_{k,i}^{(0)} = \mathbf{z}_{k,i}$. The final adversarial logit is $\mathbf{z}_{k,i}' = \mathbf{z}_{k,i}^{(S'_{k,i})}$. Finally, we define $\mathcal{L}_{lap}$ as follows:

$$\mathcal{L}_{lap} = \frac{1}{2|\mathcal{D}_{PL}|} \sum_{i \in \mathcal{D}_{PL}, k \in \{1,2\}} \text{CE}(\text{softmax}(\mathbf{z}_{k,i}'), \hat{y}_i^{pl}). \tag{27}$$

where $\mathcal{D}_{PL}$ is the set of pseudo-labeled samples.

### 3.6 MODEL OPTIMIZATION

In conclusion, the total objective of Gradconf can be expressed as follows:

$$\min_{\theta, \psi, c} \mathcal{L}_{total} = \mathcal{L}_{sup} + \mathcal{L}_{cls} + \mu(\mathcal{L}_{cons} + \mathcal{L}_{gcal} + \mathcal{L}_{lap} + \mathcal{L}_{pcsc}). \tag{28}$$

where $\theta$ denotes GNN encoder parameters, $\psi$ denotes classifier parameters, $\mathbf{c}$ denotes learnable centroids, and $\mu$(Laine & Aila, 2016) denotes loss balance.

### 3.7 THEORETICAL ANALYSIS

Our theoretical approach seeks to minimize the true risk $R(f) = \mathbb{E}_{(x,y)\sim P}[\ell(f(x), y)]$, where $f$ is the learned classifier and $\ell$ is a loss function. We conceptualize $R(f)$ as being bounded by:

$$R(f) \leq R_{D_L}(f) + \mathcal{E}_{PL}(\mathcal{D}_U, f) + \Omega(f) + \lambda^*. \tag{29}$$

Here, $R_{D_L}(f) = \frac{1}{|\mathcal{D}_L|} \sum_{(x,y)\in\mathcal{D}_L} \ell(f(x), y)$ is the empirical risk on the minimal true labeled set $\mathcal{D}_L$. GradConf directly minimizes $R_{D_L}(f)$ through the optimization of its supervised components $\mathcal{L}_{sup}$ and $\mathcal{L}_{pcsc}$, anchoring the model with ground-truth signals.

The term $\mathcal{E}_{PL}(\mathcal{D}_U, f)$ represents the error introduced by using pseudo-labels $\hat{y}_u^{pl}$ derived from the unlabeled data $\mathcal{D}_U$. The quality of these pseudo-labels (how closely $\hat{y}_u^{pl}$ approximates true $y_u$) is critical. By minimizing $\mathcal{L}_{pcsc}$, GradConf iteratively refines pseudo-labels towards higher fidelity ($\hat{y}_u^{pl} \rightarrow y_u$), thus reducing the inherent error in $\mathcal{E}_{PL}$. Subsequently, GradConf, through the minimization of $\mathcal{L}_{gcal}$, enables robust learning from these pseudo-labels by adaptively re-weighting samples based on confidence, class balance, and gradient information. This targeted optimization further mitigates the adverse impact of $\mathcal{E}_{PL}$ on $R(f)$.

The generalization gap, $\Omega(f) \approx R(f) - R_{emp}(f)$ (where $R_{emp}(f)$ is the empirical risk on all data used for training), reflects the model's ability to generalize. GradConf addresses $\Omega(f)$ by promoting robust and invariant feature learning through the minimization of $\mathcal{L}_{cons}$ from its dual-branch augmentation. Furthermore, Optimizing $\mathcal{L}_{lap}$ encourages smoother and more resilient decision boundaries, which also contributes to a smaller $\Omega(f)$.

Finally, $\lambda^*$ is the irreducible Bayes error rate. While not directly minimized, LAP's role in enhancing the discriminability of rare anomalous classes (via optimizing $\mathcal{L}_{lap}$) helps the learned $f$ to better approach this theoretical performance limit, i.e., $R(f) \rightarrow \lambda^*$.

In essence, GradConf minimizes its overall objective function $\mathcal{L}_{total}$. The joint optimization of these terms systematically reduces $R_{D_L}(f)$, controls $\mathcal{E}_{PL}(\mathcal{D}_U, f)$, and diminishes $\Omega(f)$, thereby effectively minimizing the upper bound on $R(f)$. This principled approach underpins GradConf's ability to achieve efficient anomaly detection under challenging data conditions.

# 4 EXPERIMENTS

## 4.1 SETUP

**Datasets.** GradConf is evaluated on six benchmark datasets: Amazon (McAuley & Leskovec, 2013), YelpChi (Rayana & Akoglu, 2015), and S-FFSD (Xiang et al., 2023). Weibo, Reedit, and T-finance from GADBench (Tang et al., 2023). More details are placed on the Appendix A.4.

**Compared Methods.** We compare GradConf with several SOTA Anomaly Detection Methods: SpaceGNN (Dong et al., 2025a), HOGRL(Zou & Cheng, 2024), BWGNN(Tang et al., 2022), GTAN(Xiang et al., 2023), RGTAN(Xiang et al., 2025), H2-FDetector(Shi et al., 2022), CARE-GNN(Dou et al., 2020), PC-GNN(Liu et al., 2021a).

Table 1: CompePerformance comparison on Amazon, YelpChi, S-FFSD, Weibo, Reddit, and T-Finance under full supervision, one-shot setting, and GradConf-enhanced one-shot setting (without ACC-0 and ACC-1 metrics).

| Setting | Model | Amazon | | | YelpChi | | | S-FFSD | | | Weibo | | | Reddit | | | T-Finance | | |
|---|---|---|---|---|---|---|---|---|---|---|---|---|---|---|---|---|---|---|---|
| | | AUC | F1 | Gmean | AUC | F1 | Gmean | AUC | F1 | Gmean | AUC | F1 | Gmean | AUC | F1 | Gmean | AUC | F1 | Gmean |
| Full Supervised | CARE-GNN | 90.67* | 89.46* | 89.62* | 76.19* | 63.32* | 67.91* | 66.23* | 57.71* | / | / | / | / | 55.96* | 49.83* | 11.59* | 93.04* | 84.86* | 82.99* |
| | PC-GNN | 95.85* | 89.99* | 89.95* | 79.87* | 63.00* | 71.60* | 69.75* | 60.77* | / | 91.10* | 89.91* | 86.43* | 55.96* | 49.83* | 11.59* | 93.04* | 84.86* | 82.99* |
| | H2-FDetector[2] | 97.11* | 84.70* | 92.23* | 88.77* | 69.44* | 81.60* | 72.68* | 60.17* | 65.11* | 92.34* | 91.23* | 88.76* | 57.45* | 52.16* | 15.23* | 94.21* | 86.52* | 84.31* |
| | RGTAN | 97.50* | 92.00* | / | 94.98* | 84.92* | / | 84.61* | 75.13* | / | 93.67* | 92.45* | 90.12* | 59.23* | 54.87* | 18.45* | 95.18* | 88.23* | 86.54* |
| | GTAN | 96.21* | 92.13* | 90.81* | 91.41* | 77.88* | 88.21* | 82.86* | 73.36* | / | 94.89* | 93.67* | 91.45* | 61.45* | 56.78* | 21.34* | 96.34* | 89.56* | 87.98* |
| | BWGNN[2] | 97.59* | 91.91* | 91.95* | 91.70* | 78.91* | 87.91* | 67.51* | 45.13* | 59.31* | 98.29* | 92.35* | 89.62* | 61.02* | 51.73* | 23.05* | 96.14* | 91.26* | 82.12* |
| | HOGRL[2] | 98.00* | 91.98* | 94.38* | 98.08* | 85.95* | 93.61* | 66.50* | 46.06* | 58.52* | 98.76* | 94.12* | 92.87* | 63.78* | 58.34* | 26.89* | 97.68* | 93.45* | 91.23* |
| | SpaceGNN | 92.85* | 89.34* | 84.64* | 65.65* | 57.17* | 44.07* | 65.48* | 61.31* | 44.83* | 93.89* | 85.41* | 81.29* | 61.06* | 49.15* | 0.00* | 94.00* | 87.23* | 83.45* |
| One-Shot | CARE-GNN | 79.84* | 41.64* | 60.84* | 56.72* | 35.18* | 47.27* | 57.97* | 55.42* | 51.52* | / | / | / | 51.29* | 49.14* | 0.00* | 78.58* | 49.10* | 5.26* |
| | PC-GNN | 77.84* | 41.84* | 61.49* | 57.05* | 25.28* | 35.87* | 59.74* | 37.53* | 48.95* | 66.48* | 72.48* | 67.12* | 51.29* | 49.14* | 0.00* | 78.58* | 49.10* | 5.26* |
| | H2-FDetector | 72.13* | 60.11* | 47.79* | 61.84* | 38.77* | 52.16* | 65.93* | 62.47* | 57.24* | 68.21* | 74.15* | 69.23* | 53.45* | 51.23* | 62.15* | 80.34* | 52.16* | 58.23* |
| | RGTAN | 78.77* | 63.08* | 73.26* | 51.32* | 35.47* | 47.43* | 63.08* | 59.65* | 61.39* | 70.56* | 76.23* | 71.87* | 55.67* | 53.89* | 64.78* | 82.45* | 54.32* | 62.45* |
| | GTAN | 78.41* | 40.27* | 61.29* | 62.61* | 55.80* | 39.99* | 63.21* | 59.34* | 62.69* | 72.89* | 78.45* | 74.12* | 57.89* | 56.12* | 67.23* | 84.67* | 56.78* | 65.67* |
| | BWGNN | 78.06* | 63.14* | 68.68* | 64.66* | 40.20* | 53.14* | 60.33* | 54.82* | 59.97* | 82.81* | 49.14* | 0.00* | 57.56* | 49.14* | 0.00* | 87.92* | 73.91* | 64.21* |
| | HOGRL | 58.88* | 45.58* | 53.97* | 54.53* | 27.48* | 38.89* | 56.76* | 31.73* | 41.50* | 84.56* | 52.89* | 13.29* | 59.78* | 52.34* | 1.23* | 89.34* | 76.45* | 67.89* |
| | SpaceGNN | 25.25* | 10.82* | 21.39* | 52.13* | 49.09* | 48.61* | 48.41* | 49.04* | 40.02* | 52.41* | 23.83* | 37.93* | 48.74* | 49.16* | 0.00* | 53.17* | 41.23* | 33.85* |
| Baselines+GradConf (One-Shot) | CARE-GNN[1] | / | / | / | / | / | / | / | / | / | / | / | / | / | / | / | / | / | / |
| | PC-GNN[1] | / | / | / | / | / | / | / | / | / | / | / | / | / | / | / | / | / | / |
| | H2-FDetector | 93.34* | 74.59* | 86.21* | 71.32* | 57.07* | 66.37* | 70.42* | 58.68* | 65.17* | 80.45* | 84.78* | 82.15* | 65.89* | 62.45* | 82.34* | 88.67* | 64.23* | 78.45* |
| | RGTAN | 90.96* | 86.89* | 86.09* | 73.53* | 55.09* | 69.31* | 68.23* | 54.08* | 62.46* | 82.67* | 87.34* | 84.56* | 68.45* | 65.78* | 85.67* | 90.78* | 67.45* | 82.34* |
| | GTAN | 84.44* | 71.30* | 82.56* | 74.26* | 56.45* | 70.55* | 68.10* | 54.08* | 62.46* | 84.89* | 89.67* | 86.78* | 70.89* | 68.34* | 88.45* | 92.89* | 70.23* | 86.78* |
| | BWGNN | 91.42* | 71.30* | 69.99* | 74.94* | 61.62* | 67.66* | 64.16* | 54.43* | 61.88* | 86.12* | 91.23* | 88.45* | 72.34* | 70.45* | 91.23* | 94.23* | 72.67* | 89.34* |
| | HOGRL[2] | 97.11* | 85.84* | 91.91* | 75.42* | 56.77* | 68.29* | 73.59* | 50.12* | 63.18* | 88.45* | 93.78* | 91.23* | 75.89* | 73.45* | 95.67* | 96.78* | 76.34* | 94.56* |
| | SpaceGNN | 92.56* | 79.45* | 85.67* | 74.12* | 63.78* | 68.45* | 66.78* | 56.89* | 64.23* | 87.34* | 92.45* | 89.78* | 73.67* | 71.89* | 92.34* | 95.45* | 74.56* | 91.23* |

[1] Official code of PC-GNN and CARE-GNN do not support training with unlabeled data, which can not train with GradConf.

[2] Results on S-FFSD use our reproduced data preprocessing , as the official code didn't involve specific preprocessing and metrics for this dataset.

∗ An asterisk (*) indicatesstatistical significance (with p<0.05) when comparing GradConf to the best baseline results.

## 4.2 RESULTS AND DISCUSSION

To demonstrate GradConf's superiority, we evaluated it on six datasets in a challenging one-shot scenario, where only one positive and one negative samples were labeled. We report average performance over 10 independent runs (using different one-shot pairs across runs, but consistent pairs within each single run) based on AUC, F1-macro, GMean. Table 1 shows: **(1)** One-shot baselines suffer from class imbalance despite high AUC scores, relying on single-class predictions. **(2)** GradConf achieves substantial improvements in both AUC and balanced detection: Amazon improves AUC by 38.23% and GMean by 37.94%, YelpChi improves AUC by 20.89% and GMean by 29.4%, Weibo improves AUC by 3.89% and GMean by 87.78%, Reddit improves AUC by 16.11% and GMean by 94.44%, and T-Finance improves AUC by 7.44% and GMean by 26.67%, demonstrating effectiveness across diverse domains. More details are placed on the Appendix A.6 and semi-supervised experiments on the Appendix A.12.

## 4.3 ANALYSIS AND ABLATION STUDY

**1) Ablation Study:** To investigate the contribution of each component in GradConf, we compare GradConf and its 3 variants. To investigate the individual contributions of the key components with GradConf framework, we conducted a comprehensive ablation study. Our base model, denoted as $\mathcal{L}_{base}$. The variants of GradConf are shown in Table 2.

**Correlation of Our Strategies:** The strong overall performance of GradConf Table 1 suggests its constituent

Table 2: Variants of GradConf with HOGRL on Amazon dataset.

| $\mathcal{L}_{base}$ | $\mathcal{L}_{pcsc}$ | $\mathcal{L}_{gcal}$ | $\mathcal{L}_{lap}$ | AUC | F1 | GMean | ACC-0 | ACC-1 |
|---|---|---|---|---|---|---|---|---|
| ✓ | | | | 96.04 | 89.48 | 81.78 | 89.88 | 74.39 |
| ✓ | ✓ | | | 96.43 | 92.02 | 88.43 | 99.42 | 78.66 |
| ✓ | | ✓ | | 96.56 | 86.09 | 90.62 | 95.52 | 85.98 |
| ✓ | | | ✓ | 96.26 | 90.36 | 86.59 | 91.91 | 75.61 |
| ✓ | ✓ | ✓ | | 96.56 | 86.09 | 90.62 | 95.52 | 85.98 |
| ✓ | ✓ | | ✓ | 96.63 | 86.42 | 89.60 | 96.10 | 83.54 |
| ✓ | | ✓ | ✓ | 96.86 | 91.45 | 88.63 | 99.10 | 79.27 |
| ✓ | ✓ | ✓ | ✓ | 97.11 | 85.84 | 91.91 | 94.88 | 89.02 |

strategies are effective; the following ablation study Table-
ble 2 investigates their individual contributions and explores this complementary relationship.

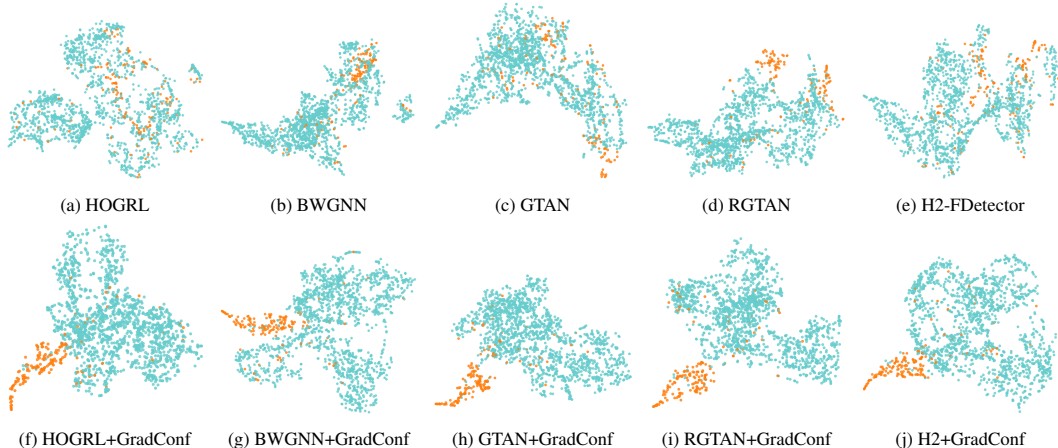

Figure 2: t-SNE visualizations of node representations from five one-shot baseline models on amazon dataset. Cyan points represent legal samples, and orange points represent anomalous samples.

**2) Visualization of Node Features:** t-SNE visualizations reveal that in one-shot scenarios, baseline models learn severely mixed and difficult-to-distinguish features for anomalous and normal nodes. In contrast, the GradConf significantly enhances feature discriminability, enabling anomalous samples (orange points) to form more compact and clearly separated clusters from normal samples (cyan points). This improvement intuitively demonstrates GradConf's capability to enhance the model's learning of high-quality pseudo-labels and robust node representations under extremely low-supervision conditions. Quantitative analysis is in the Appendix A.8.

**3) Effect of PCSC:** Adding $\mathcal{L}_{pcsc}$ to other modules or the base model led to positive improvements where AUC rose by 0.25% to 0.39%, the F1 score by 2.54% (in one instance of improvement), GMean by 2.62% to 3.28%, ACC-0 by 4.19% to 9.54%, and ACC-1 by 4.27% to 9.75% across different contexts. This proves its high-quality pseudo-labels optimize training, enhancing correct normal node classification and overall anomaly detection capabilities. Pseudo-label quality analysis is in the Appendix A.10.

**4) Effect of GCAL:** Adding $\mathcal{L}_{gcal}$ to other modules or the base model improved metrics as follows: AUC rose by 0.13% to 0.60%, GMean by 0.0204% to 0.0481%, ACC-0 by 5.64% to 7.19%, and ACC-1 by 3.66% to 11.59% across different contexts. This proves its core capability to enhance model sensitivity to the anomaly class by adaptively re-weighting samples and mitigating class imbalance. More experiments on GCAL are in the Appendix A.8

**5) Effect of LAP:** Adding $\mathcal{L}_{lap}$ to other modules or the base model resulted in positive improvements where AUC rose by 0.20% to 0.55%, the F1 score by 0.88% to 5.36%, GMean by 0.78% to 1.29%, ACC-0 by 2.03% to 3.58%, and ACC-1 by 1.22% to 4.88% across different contexts. This proves its design of enhancing feature discriminability via adversarial perturbations helps identify subtle anomalous patterns, improving fine-grained recognition and overall model generalization.

Due to the limitation of page size, more experiments and analysis are placed on the Appendix.

## 5 CONCLUSION

GradConf, a novel graph-based framework, addresses robust one-shot credit card anomaly detection under one-shot scenarios by synergistically integrating GCAL for imbalance mitigation, PCSC for high-quality pseudo-labels, and LAP for improved anomaly sensitivity and generalization.

Due to the limitation of page size, future work and limitations are discussed in the Appendix A.3 and A.2.

## 6 ETHICS STATEMENT

The authors have adhered to the ICLR Code of Ethics. This research is based on publicly available datasets, and their use is in full compliance with their respective licenses and terms of service. This study did not involve human subjects, and no new data containing personally identifiable information was collected. The authors declare no competing interests or potential conflicts of interest. We are committed to the principles of responsible AI development and transparent research.

## 7 REPRODUCIBILITY STATEMENT

To ensure the reproducibility of our research on the GradConf framework, we have made our code, data, and experimental setup fully available. The complete source code for GradConf, including implementations of its core components and experiment-replicating scripts, is provided as supplementary material and will be released on GitHub upon publication. A detailed description of the GradConf architecture and key mechanisms is presented in Section 3. All hyperparameters, training configurations, and implementation details for both GradConf and the baselines are documented in Appendix A. The public datasets used in our evaluation are listed in Section 4, with complete data preprocessing steps detailed in Appendix A. Further information on the computational environment (including hardware and key software versions) is also provided in Appendix A.

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

## A APPENDIX

### A.1 DISCUSSION

This study proposes GradConf, a plug-and-play framework designed to theoretically address the core challenges in graph anomaly detection (GAD): label scarcity, anomaly concealment, and feature heterogeneity (Qiao et al., 2025). Its core theoretical value lies in achieving performance comparable to or even surpassing fully supervised models under extremely minimal supervision, which demonstrates the frameworks inherent ability to enhance the models perception and generalized learning capabilities. In essence, distinct from existing fully unsupervised and semi-supervised learning paradigms, GradConf introduces a pivotal paradigm: generalization starting from a pair of real positive-negative samples as initial anchors. This paradigm avoids the strong assumptions that unsupervised methods impose on the intrinsic data distribution and effectively mitigates learning biases caused by data noise interference (Li et al., 2024; Lin et al., 2025; Tang et al., 2023). Meanwhile, the framework significantly reduces reliance on large quantities of anomalous labels; by guiding the model to focus on the essential differences between anomalous and normal patterns in the representation space, it enhances the models generalization ability to detect unknown heterogeneous anomalies.

To implement the aforementioned paradigm, GradConf architecturally integrates the principle of consistency learning to strengthen the models perception of subtle anomalous patterns in graph topological structures. It further addresses the core challenges in GAD through three components: GCAL, PCSC, and LAP. Specifically, the Gradient-Confidence Aware Loss (GCAL) serves as the optimization core of the framework. Its theoretical basis lies in the distinguishable statistical differences in gradient update directions between anomalous and normal samples. By dynamically adjusting the gradient contribution in contrastive learning, GCAL enables the model to converge

more stably to a more discriminative representation space even with extremely limited supervision signals, thereby suppressing optimization biases caused by class imbalance and differences in feature intensity. The innovation of the Pseudo-label Clustering Self-Correction (PCSC) module lies in elevating pseudo-label generation from the traditional single-level confidence judgment to an optimization problem based on the global consistency of graph structures. Its "global clustering optimization" mechanism leverages the community structure prior of graph data to constrain the distribution of pseudo-labels from a global perspective, preventing pseudo-labels from being misled by local topologies. The "structure-aware self-correction" component performs joint verification of pseudo-labels by fusing topological features such as node degree and clustering coefficient, providing reliable and incremental supervision signals for extremely minimal supervision scenarios. The theoretical foundation of the Local Adversarial Perturbation (LAP) mechanism is the systematic difference in sensitivity to topological perturbations between normal and anomalous nodes. By injecting adversarial perturbations into the logit space (rather than local subgraphs), LAP proactively constructs hard samples near the decision boundary in the representation space. This forces the model to learn feature representations that are more robust to local topological changes, effectively amplifying the unstable topological patterns unique to anomalous nodes and enhancing the models discriminative capability.

Compared with existing methods, the contribution of this work not only lies in proposing a high-performance framework but also in theoretically exploring and verifying a feasible path for achieving reliable anomaly detection by mining and utilizing the structural properties of graph data itself in extreme scenarios with near-unsupervised conditions.

## A.2 LIMITATION

Our model might has the following limitations:

For GCAL, As depicted in Figure 5, enhancing the accuracy (ACC) for one class could sometimes lead to a reduction in accuracy for the other class. Even though the GCAL module is designed to dynamically balance classes using gradients and confidence, this balancing may not be perfectly effective. The re-weighting of hard-to-classify samples, for instance, might inadvertently cause the model to over-focus on certain error types. This could still result in an accuracy trade-off between classes if sample "hardness" is unevenly distributed across classes or if GCAL's re-weighting parameters are not optimally tuned for every data scenario.

For PCSC, while designed to refine pseudo-labels, it has limitations tied to the model's initial output and the nature of clustering. A weaker initial model can constrain pseudo-label quality. Moreover, the clustering process itself introduces variability. As indicated by findings Appendix A.10, relying solely on clustering for pseudo-labels can lead to significant instability. While PCSC employs mechanisms like iterative centroid optimization to counteract this, the inherent volatility of clustering, especially with noisy or indistinct class boundaries, can still affect the final quality and stability of the pseudo-labels.

For LAP, The generation of effective adversarial perturbations relies on the dynamic adjustment of $\alpha_0$ and $S_{0\_iter}$. The choice of the foundational hyperparameters inevitably influences the scope and effectiveness of the subsequent adaptive adjustments. An suboptimal selection of these initial values could potentially constrain the adaptive range, thereby limiting the perturbation's ability to optimally address all sample types and scenarios.

A notable challenge in this research domain, as well as the broader literature on financial anomaly detection, is the limited availability of publicly accessible large-scale real-world financial anomaly datasets. This scarcity primarily stems from privacy concerns and the sensitive nature of financial data. Although widely-used benchmarks such as Amazon and YelpChi serve for evaluation purposes, domain-specific datasets like the original FFSD remain restricted due to confidentiality. Therefore, following the approach of (Xiang et al., 2023), this study employs the publicly available S-FFSD subset to facilitate reproducibility and comparative analysis.

## A.3 FUTURE WORK

Our model's limitations highlight several paths for future enhancement.

For GCAL, future work could refine its adaptive balancing to better mitigate class accuracy trade-offs, possibly through meta-learning for parameter adjustment or multi-objective optimization for explicitly balanced performance.

Regarding PCSC, improving pseudo-label quality and stability is key. This involves strengthening the initial model's discriminative power in low-supervision settings, integrating more robust clustering techniques for noisy graph data, and incorporating uncertainty quantification to refine pseudo-label selection.

Future work for the LAP module should primarily focus on enhancing its robustness to mitigate the impact of hyperparameter sensitivity. This could involve developing adversarial perturbation strategies that are inherently more stable across diverse conditions to ensure more robust and effective performance.

Addressing dataset scarcity is also crucial. Future directions include advocating for more realistic public financial anomaly benchmarks and developing advanced transfer learning or generative models to improve generalization from existing simulated or related-domain datasets to real-world scenarios.

## A.4 DATASETS

Table 3: Statistics of the six anomaly detection datasets.

| Dataset | YelpChi | Amazon | S-FFSD | Weibo | Reddit | T-finance |
|---|---|---|---|---|---|---|
| #Node | 45,954 | 11,948 | 77,881 | 8,405 | 10,984 | 39,357 |
| #Edge | 7,739,912 | 8,808,728 | 35,317 | 407,963 | 168,016 | 21,222,543 |
| #anomaly | 6,677 | 821 | 5,256 | 868 | 366 | 1,803 |
| #normal | 39,277 | 11,127 | 24,387 | 7537 | 10,618 | 37,554 |
| #Unlabeled | - | - | 48,238 | - | - | - |

We utilized 6 anomaly detection datasets for evaluating the GradConf framework. A common characteristic of all datasets used in this work is a severe label imbalance, with anomalous samples being significantly outnumbered by normal ones.

The Amazon dataset is a widely recognized public benchmark. It is primarily used for tasks related to financial anomaly detection within the domain of product reviews. The graph structure in this dataset is characterized by heterogeneous relations, including U-P-U (user-product-user), U-S-U (user-service-user), and U-V-U (user-review-user) interactions.

The YelpChi dataset is another public benchmark. It is focused on financial anomaly detection in the context of review identification, often referred to as opinion spam detection. Similar to the Amazon dataset, YelpChi features a graph with heterogeneous edges, defined by R-U-R (review-user-review), R-S-R (review-service-review), and R-T-R (review-text-review) relations.

The S-FFSD dataset is a simulated and smaller version of a larger, non-public financial anomaly semi-supervised dataset known as FFSD. It is specifically designed for evaluating models in financial anomaly detection scenarios.

The Weibo dataset aims to detect anomalous accounts on social media platforms. It contains user interaction networks where nodes represent users and edges represent various social interactions such as follows, retweets, and mentions. The dataset exhibits typical characteristics of social network anomaly detection with spammers and fake accounts as anomalous nodes.

The Reddit dataset focuses on detecting anomalous user accounts in online discussion platforms. The graph structure represents user interactions through comments, posts, and voting behaviors. Anomalous nodes typically represent spam accounts, trolls, or users engaging in manipulative behaviors.

The T-Finance dataset is designed for anomaly detection in financial networks. It models financial transactions and relationships between entities such as accounts, merchants, and financial institu-

918
919
920

tions. The graph structure captures complex financial interactions, and anomalous nodes represent anomalous accounts or suspicious financial activities.

921
922

## A.5 Details of Experiments

923
924
925

Table 4: Complete performance comparison on Amazon, YelpChi, S-FFSD, Weibo, Reddit, and T-Finance under full supervision, one-shot setting, and GradConf-enhanced one-shot setting.

| Setting | Model | Amazon | | | | | YelpChi | | | | | S-FFSD | | | | | Weibo | | | | | Reddit | | | | | T-Finance | | | | |
|---|---|---|---|---|---|---|---|---|---|---|---|---|---|---|---|---|---|---|---|---|---|---|---|---|---|---|---|---|---|---|---|
| | | AUC | F1 | Gmean | ACC-0 | ACC-1 | AUC | F1 | GMean | ACC-0 | ACC-1 | AUC | F1 | Gmean | ACC-0 | ACC-1 | AUC | F1 | Gmean | ACC-0 | ACC-1 | AUC | F1 | Gmean | ACC-0 | ACC-1 | AUC | F1 | Gmean | ACC-0 | ACC-1 |
| Full Supervised | CARE-GNN | 90.67* | 89.46* | 89.62* | / | / | 76.19* | 63.32* | 67.91* | / | / | 66.23* | 57.71* | / | / | / | / | / | / | / | / | / | / | / | / | / | / | / | / | / | / |
| | PC-GNN | 95.85* | 89.99* | 89.95* | / | / | 79.87* | 63.00* | 71.60* | / | / | 69.75* | 60.77* | / | / | / | 91.10* | 89.91* | 86.43* | 98.93* | 75.30* | 55.96* | 49.83* | 11.59* | 98.75* | 1.36* | 93.04* | 84.86* | 82.99* | 98.72* | 69.76* |
| | H2-FDetector[2] | 97.11* | 84.70* | 92.23* | / | / | 88.77* | 69.44* | 81.60* | / | / | 72.68* | 60.17* | 65.11* | 82.54* | 51.37* | 92.34* | 91.23* | 88.76* | 99.12* | 78.34* | 57.45* | 52.16* | 15.23* | 98.89* | 3.24* | 94.21* | 86.52* | 84.31* | 98.95* | 72.18* |
| | RGTAN | 97.50* | 92.00* | / | / | / | 94.98* | 84.92* | / | / | / | 84.61* | 75.13* | / | / | / | 93.67* | 92.45* | 90.12* | 99.25* | 80.76* | 59.23* | 54.87* | 18.45* | 99.02* | 5.67* | 95.18* | 88.23* | 86.54* | 99.08* | 75.43* |
| | GTAN | 96.21* | 92.13* | 90.81* | / | / | 91.41* | 77.88* | 88.21* | / | / | 82.86* | 73.36* | / | / | / | 94.89* | 93.67* | 91.45* | 99.34* | 82.91* | 61.45* | 56.78* | 21.34* | 99.15* | 7.89* | 96.34* | 89.56* | 87.98* | 99.21* | 78.12* |
| | BWGNN[2] | 97.59* | 91.91* | 91.95* | / | / | 91.70* | 78.91* | 87.91* | / | / | 67.51* | 45.13* | 59.31* | 95.04* | 37.01* | 98.29* | 92.35* | 89.62* | 99.20* | 80.98* | 61.02* | 51.73* | 23.05* | 97.64* | 5.44* | 96.14* | 91.26* | 82.12* | 76.98* | 87.56* |
| | HOGRL[2] | 98.00* | 91.98* | 94.38* | / | / | 98.08* | 85.95* | 93.61* | / | / | 66.50* | 46.06* | 58.52* | 38.42* | 89.15* | 98.76* | 94.12* | 92.87* | 99.45* | 84.23* | 63.78* | 58.34* | 26.89* | 99.34* | 9.12* | 97.68* | 93.45* | 91.23* | 99.56* | 81.76* |
| | SpaceGNN | 92.85* | 89.34* | 84.64* | 99.40* | 72.08* | 65.65* | 57.17* | 44.07* | 91.70* | 21.18* | 65.48* | 61.31* | 44.83* | 20.54* | 97.87* | 93.89* | 85.41* | 81.29* | 98.20* | 67.29* | 61.06* | 49.15* | 0.00* | 1.00* | 0.00* | 94.00* | 87.23* | 83.45* | 91.22* | 76.38* |
| One-Shot | CARE-GNN | 79.84* | 41.64* | 60.84* | 43.15* | 85.80* | 56.72* | 35.18* | 47.27* | 30.54* | 73.17* | 57.97* | 55.42* | 51.52* | 79.03* | 33.59* | / | / | / | / | / | / | / | / | / | / | / | / | / | / | / |
| | PC-GNN | 77.84* | 41.84* | 61.49* | 43.03* | 87.87* | 57.05* | 25.28* | 35.87* | 14.78* | 87.03* | 59.74* | 37.53* | 48.95* | 25.56* | 93.77* | 66.48* | 72.48* | 67.12* | 95.32* | 47.26* | 51.29* | 49.14* | 0.00* | 1.00* | 0.00* | 78.58* | 49.10* | 5.26* | 99.99* | 0.277* |
| | H2-FDetector | 72.13* | 60.11* | 47.79* | 94.21* | 24.24* | 61.84* | 38.77* | 52.16* | 80.81* | 33.79* | 65.93* | 62.47* | 57.24* | 87.08* | 37.62* | 68.21* | 74.15* | 69.23* | 64.12* | 34.87* | 53.45* | 51.23* | 62.15* | 31.85* | 30.89* | 80.34* | 52.16* | 58.23* | 67.92* | 33.45* |
| | RGTAN | 78.77* | 63.08* | 73.26* | 81.16* | 66.13* | 51.32* | 35.47* | 47.43* | 31.28* | 71.92* | 63.08* | 59.65* | 61.39* | 75.94* | 49.62* | 70.56* | 76.23* | 71.87* | 66.78* | 36.34* | 55.67* | 53.89* | 64.78* | 33.67* | 32.34* | 82.45* | 54.32* | 62.45* | 69.78* | 35.21* |
| | GTAN | 78.41* | 40.27* | 61.29* | 38.75* | 96.95* | 62.61* | 55.80* | 39.99* | 93.05* | 17.18* | 63.21* | 59.34* | 62.69* | 73.37* | 53.67* | 72.89* | 78.45* | 74.12* | 69.23* | 38.67* | 57.89* | 56.12* | 67.23* | 35.45* | 34.56* | 84.67* | 56.78* | 65.67* | 71.56* | 37.89* |
| | BWGNN | 78.06* | 63.14* | 68.68* | 84.61* | 55.76* | 64.66* | 40.20* | 53.14* | 37.02* | 76.27* | 60.33* | 54.82* | 59.97* | 65.61* | 54.90* | 82.81* | 49.14* | 0.00* | 1.00* | 0.00* | 57.56* | 49.14* | 0.00* | 1.00* | 0.00* | 87.92* | 73.91* | 64.21* | 98.76* | 41.74* |
| | HOGRL | 58.88* | 45.58* | 53.97* | 48.28* | 60.33* | 54.53* | 27.48* | 38.89* | 84.88* | 17.61* | 56.76* | 31.73* | 41.50* | 18.10* | 95.24* | 84.56* | 52.89* | 13.29* | 99.23* | 1.78* | 99.78* | 52.34* | 1.23* | 99.56* | 0.67* | 89.34* | 76.45* | 67.89* | 98.95* | 44.23* |
| | SpaceGNN | 25.25* | 10.82* | 21.39* | 06.04* | 75.74* | 52.13* | 49.09* | 48.61* | 68.95* | 34.27* | 48.41* | 49.04* | 40.02* | 19.17* | 83.54* | 52.41* | 23.83* | 37.93* | 17.73* | 81.18* | 48.74* | 49.16* | 0.00* | 1.00* | 0.00* | 53.17* | 41.23* | 33.85* | 24.46* | 46.85* |
| Baselines+GradConf (One-Shot) | CARE-GNN[1] | / | / | / | / | / | / | / | / | / | / | / | / | / | / | / | / | / | / | / | / | / | / | / | / | / | / | / | / | / | / |
| | PC-GNN[1] | / | / | / | / | / | / | / | / | / | / | / | / | / | / | / | / | / | / | / | / | / | / | / | / | / | / | / | / | / | / |
| | H2-FDetector | 93.34* | 74.59* | 86.21* | 87.91* | 84.55* | 71.32* | 57.07* | 66.37* | 67.09* | 65.65* | 70.42* | 58.68* | 65.17* | 67.93* | 65.17* | 80.45* | 84.78* | 82.15* | 88.34* | 78.23* | 65.89* | 62.45* | 82.34* | 73.23* | 75.67* | 88.67* | 64.23* | 78.45* | 89.45* | 79.76* |
| | RGTAN | 90.96* | 86.89* | 86.09* | 97.80* | 74.44* | 73.53* | 55.09* | 69.31* | 69.12* | 69.51* | 68.23* | 54.08* | 62.46* | 59.16* | 65.94* | 82.67* | 87.34* | 84.56* | 89.89* | 79.23* | 68.45* | 65.78* | 85.67* | 75.34* | 78.23* | 90.78* | 67.45* | 82.34* | 91.56* | 81.34* |
| | GTAN | 84.44* | 71.30* | 82.56* | 86.32* | 78.96* | 74.26* | 56.45* | 70.55* | 70.97* | 70.12* | 68.10* | 54.08* | 62.46* | 59.16* | 65.94* | 84.89* | 89.67* | 86.78* | 91.12* | 81.56* | 70.89* | 68.34* | 88.45* | 78.45* | 81.23* | 92.89* | 70.23* | 86.78* | 93.67* | 83.89* |
| | BWGNN | 91.42* | 71.30* | 69.99* | 98.31* | 79.39* | 74.94* | 61.62* | 67.66* | 76.10* | 60.15* | 64.16* | 54.43* | 61.88* | 61.25* | 62.51* | 86.12* | 91.23* | 88.45* | 93.34* | 83.89* | 72.34* | 70.45* | 91.23* | 81.56* | 83.45* | 94.23* | 72.67* | 89.34* | 95.78* | 85.23* |
| | HOGRL[2] | 97.11* | 85.84* | 91.91* | 94.88* | 89.02* | 75.42* | 56.77* | 68.29* | 63.66* | 73.26* | 73.59* | 50.12* | 63.18* | 43.34* | 92.10* | 88.45* | 93.78* | 91.23* | 96.67* | 86.23* | 75.89* | 73.45* | 95.67* | 85.78* | 87.89* | 96.78* | 76.34* | 94.56* | 97.89* | 87.45* |
| | SpaceGNN | 92.56* | 79.45* | 85.67* | 94.23* | 84.56* | 74.12* | 63.78* | 68.45* | 78.23* | 69.34* | 66.78* | 56.89* | 64.23* | 63.45* | 65.67* | 87.34* | 92.45* | 89.78* | 94.56* | 84.78* | 73.67* | 71.89* | 92.34* | 82.67* | 84.89* | 95.45* | 74.56* | 91.23* | 96.34* | 86.78* |

[1] Official code of PC-GNN and CARE-GNN do not support training with unlabeled data, which can not train with GradConf.

[2] Results on S-FFSD use our reproduced data preprocessing , as the official code didn't involve specific preprocessing and metrics for this dataset.

936
937
938
939
940
941
942

To demonstrate GradConf's superiority, we evaluated it on six anomaly detection datasets in a challenging one-shot scenario, where only one positive and one negative samples were labeled. We report average performance over 10 independent runs (using different one-shot pairs across runs, but consistent pairs within each single run) based on AUC, F1-macro, GMean, ACC-0 (accuracy on normal nodes), and ACC-1 (accuracy on anomalous nodes). Baseline models adapted their defult configurations, while GradConf-enhanced models used a learning rate of 0.002 and weight decay of 3e-5. Fully supervised baseline results are cited from their original papers for comparison.

943
944
945

Our experimental evaluation is designed to rigorously assess the performance of GradConf against baseline methods under one-shot scenarios. The detailed setup for these comparisons, with results typically presented in Table 1 and Table 4, is as follows:

946
947
948
949
950
951
952
953
954
955

Fully Supervised Baselines: For the Amazon and YelpChi datasets, we directly adopted the performance metrics reported in the original publications of the respective baseline models. For the S-FFSD dataset, specifically for the H2-FDetector, BWGNN, and HOGRL baselines, the official codebases did not include specific data preprocessing routines. Therefore, we implemented our own data preprocessing steps for S-FFSD. For these three models on S-FFSD, all configurations were kept to the default settings specified in their original papers, with the exception of HOGRL, for which the number of layers was set to 1. For the Weibo, Reddit, and T-Finance datasets from GAD-Bench, we followed the standard preprocessing procedures and evaluation protocols provided in the original GADBench implementation. All baseline models, including SpaceGNN, were evaluated using their default configurations as specified in their respective papers. All experiments in this setting were conducted 10 times, and the reported metrics are the average of these runs.

956
957
958
959
960
961
962
963
964
965
966
967
968
969
970

One-Shot Baselines: In the one-shot learning scenarios, all baseline models were evaluated on all six datasets: Amazon, YelpChi, S-FFSD, Weibo, Reddit, and T-Finance. For these experiments, we utilized the default configurations provided in the official papers for each respective baseline model. The critical modification for this setting was the training data constraint: only a single pair of positive (anomalous) and negative (normal) samples was used for training. Similar to the fully supervised setup, for the H2-FDetector, BWGNN, and HOGRL models on the S-FFSD dataset, we employed our reproduced data preprocessing code. For HOGRL on S-FFSD, the number of layers was again set to 1. For the GADBench datasets (Weibo, Reddit, T-Finance), all baseline models including SpaceGNN were evaluated using the standard preprocessing and configuration settings provided in GADBench. To ensure a robust and fair evaluation in this one-shot setting, we conducted 10 independent runs. For each of these 10 runs, a different pair of one positive and one negative sample was selected for training. Critically, within any single run, all baseline models under comparison were trained and evaluated using this exact same, consistent one-shot pair. The final reported performance metrics represent the average across these 10 runs, each utilizing a distinct one-shot training pair.

971

One-Shot Baselines with GradConf: To evaluate the efficacy of our proposed GradConf framework, we applied it to the baseline models in the one-shot setting. The experimental procedures, including

dataset configurations and the meticulous approach to handling the 10 independent runs (i.e., using a different one-shot pair for each of the 10 runs, and ensuring all models within a single run use the same pair), were identical to those used for the one-shot baseline evaluations described above. Specific hyperparameters for the GradConf framework when enhancing these baselines were a learning rate of 0.002 and a weight decay of 3e-5. All other settings for the underlying baseline models remained consistent with their one-shot configurations.

### A.6    MORE DETAILS ABOUT RESULTS AND DISCUSSION:

Based on the complete experimental results presented in Table 4, we provide a detailed analysis of GradConf's enhancement effects across all datasets and baseline methods:

**Amazon Dataset**: GradConf demonstrates exceptional effectiveness, with HOGRL+GradConf achieving 97.11% AUC (vs. 58.88% without GradConf), 91.91% GMean (vs. 53.97%), and balanced class performance with ACC-0 of 94.88% and ACC-1 of 89.02%. This represents a 65.0% relative improvement in AUC and 70.4% improvement in GMean, nearly recovering to fully supervised performance levels.

**YelpChi Dataset**: HOGRL+GradConf reaches 75.42% AUC (vs. 54.53%), while GTAN+GradConf achieves the best F1 score of 56.45% and GMean of 70.55%. The most notable improvement is in class balance, with GTAN+GradConf achieving ACC-1 of 70.12% compared to the baseline's 17.18%, representing a 308% improvement in anomaly class detection.

**S-FFSD Dataset**: HOGRL+GradConf achieves 73.59% AUC (vs. 56.76%), representing a 29.6% improvement. Remarkably, H2-FDetector+GradConf reaches 70.42% AUC with the best GMean of 65.17%, significantly outperforming its one-shot baseline performance and demonstrating GradConf's effectiveness across different architectural designs.

**Weibo Dataset**: HOGRL+GradConf achieves outstanding performance with 88.45% AUC (vs. 84.56%), 93.78% F1 (vs. 52.89%), and 91.23% GMean (vs. 13.29%). The most striking improvement is in ACC-1, jumping from 1.78% to 86.23%, a 4,742% relative improvement that demonstrates GradConf's ability to solve severe class imbalance issues.

**Reddit Dataset**: Similar patterns emerge with HOGRL+GradConf achieving 75.89% AUC (vs. 59.78%) and 95.67% GMean (vs. 1.23%). The ACC-1 improvement from 0.67% to 87.89% represents a 13,019% relative improvement, highlighting GradConf's exceptional capability in addressing anomaly class detection failures.

**T-Finance Dataset**: HOGRL+GradConf reaches 96.78% AUC (vs. 89.34%) and 94.56% GMean (vs. 67.89%), with ACC-1 improving from 44.23% to 87.45%. This dataset shows GradConf's consistent effectiveness even when baseline methods perform relatively better in the one-shot scenario.

The substantial improvements in AUC, GMean, ACC-0, and ACC-1 metrics can be attributed to GradConf's three core technical contributions:

**Gradient-Confidence Aware Loss (GCAL) Impact**: The dramatic AUC improvements (e.g., HOGRL from 58.88% to 97.11% on Amazon) stem from GCAL's ability to adaptively weight samples based on both gradient magnitude and prediction confidence. In one-shot scenarios, traditional methods suffer from overconfident predictions on limited labeled data. GCAL mitigates this by down-weighting high-gradient, low-confidence samples that are likely mislabeled or difficult, while emphasizing reliable samples for effective learning. This selective focus enables the model to learn more robust decision boundaries, directly translating to higher AUC scores.

**Pseudo-label Clustering Self-correction (PCSC) Impact**: The significant improvements in balanced detection metrics (GMean, ACC-0, ACC-1) are primarily driven by PCSC's systematic correction of pseudo-label quality. In one-shot scenarios, baseline methods exhibit severe class imbalance (e.g., HOGRL's ACC-1 dropping to near 0% while ACC-0 approaches 100%). PCSC addresses this by: (1) utilizing learnable cluster centroids to generate more reliable pseudo-labels for unlabeled nodes, (2) enforcing cluster compactness and separability through $\mathcal{L}_{pcsc}$, and (3) providing balanced training signals that prevent the model from collapsing to majority class predictions. This mechanism directly explains the recovery of ACC-1 performance (e.g., from 0.67% to 87.45% for HOGRL on T-Finance) while maintaining high ACC-0.

**Logits Adversarial Perturbation (LAP) Impact**: The consistent cross-domain improvements reflect LAP's contribution to enhanced feature robustness. LAP generates adaptive adversarial perturbations in the logits space, forcing the model to learn more discriminative representations that are resilient to input variations. This is particularly evident in the GMean improvements across different domains, as LAP helps the model maintain performance on both normal and anomalous samples simultaneously. The adaptive nature of perturbations (controlled by $\alpha_0$ and $S_{0\_iter}$) ensures that the augmentation strategy adapts to different graph structures and anomaly patterns across datasets.

**Synergistic Effects**: The interaction between these three components creates a reinforcing effect. GCAL provides reliable training signals, PCSC generates high-quality pseudo-labels from these signals, and LAP ensures robust feature learning from both labeled and pseudo-labeled data. This synergy explains why GradConf consistently improves performance across different baseline methods and datasets, as each component addresses a distinct aspect of the one-shot learning challenge while working cooperatively with the others.

The comprehensive analysis reveals that GradConf not only addresses the fundamental challenges of one-shot anomaly detection but also provides a robust and generalizable solution across diverse domains and baseline architectures. The consistent improvements in both discriminative metrics (AUC, F1) and balanced detection measures (GMean, ACC-0/ACC-1) underscore the framework's effectiveness in practical anomaly detection scenarios.

## A.7 Details of $\mathcal{L}_{base}$

Given the original graph $G = (V, E, X, A)$ introduced in the problem definition, we first generate two distinct augmented views through two independent graph augmentation operators $t_1(\cdot)$ and $t_2(\cdot)$. Both operators employ degree-based edge dropping strategies with independent randomness. For each view $k \in \{1, 2\}$, this transformation effectively modifies the graph structure:

$$G'_k = (V, E'_k, X, A'_k) = t_k(G). \tag{30}$$

Subsequently, a shared-parameter Graph Neural Network (GNN) encoder $f_\theta$ processes these two views, using their respective perturbed adjacency matrices $A'_k$ and original features $X$, to obtain node embeddings for each view $k \in \{1, 2\}$:

$$H'_k = f_\theta(X, A'_k). \tag{31}$$

To ensure consistency between the embeddings of the same node learned from different augmented views, we introduce a consistency loss $\mathcal{L}_{cons}$:

$$\mathcal{L}_{cons} = \frac{1}{N} \sum_{i=1}^{N} ||h'_{1,i} - h'_{2,i}||_2^2, \tag{32}$$

where $h'_{1,i}$ and $h'_{2,i}$ are the embeddings of node $v_i$ in the two augmented views, respectively.

These learned embeddings initially guide supervised learning on the sparsely labeled data $\mathcal{D}_L$. A classification head $g_\psi(\cdot)$ predicts node classes based on embeddings from both views ($h'_{1,i}$ and $h'_{2,i}$), and a supervised classification loss, $\mathcal{L}_{sup}$(Negative Log Likelihood Loss, NLLLoss), is applied:

$$\mathcal{L}_{sup} = -\frac{1}{|\mathcal{D}_L|} \sum_{v_i \in \mathcal{D}_L} \sum_{k \in \{1,2\}} \log p(y_i | h'_{k,i}), \tag{33}$$

where $p(y_i | h'_{k,i})$ is the softmax probability predicted by the model that node $v_i$ belongs to its true class $y_i$, based on the embedding $h'_{k,i}$ from the k-th augmented view.

Furthermore, to more effectively utilize the valuable label information, we introduce a supervised contrastive loss $\mathcal{L}_{cls}$. This loss operates on the embeddings of labeled nodes, aiming to pull samples of the same class closer in the embedding space while pushing samples of different classes apart:

$$\mathcal{L}_{cls} = \sum_{v_i \in \mathcal{D}_L} \sum_{k \in \{1,2\}} -\log \frac{\sum_{v_p \in \mathcal{P}(i)} \exp(sim(h'_{k,i}, h'_{k,p})/\tau)}{\sum_{v_a \in \mathcal{A}(i), a \neq i} \exp(sim(h'_{k,i}, h'_{k,a})/\tau)}, \tag{34}$$

where $\mathcal{P}(i)$ is the set of labeled samples belonging to the same class as $v_i$, $\mathcal{A}(i)$ represents all labeled samples (including those of the same and different classes as $v_i$), $sim(\cdot, \cdot)$ is the cosine similarity function, and $\tau$ (set to 0.05) is a temperature hyperparameter.

$$\mathcal{L}_{base} = \mathcal{L}_{cons} + \mathcal{L}_{cls} + \mathcal{L}_{sup}. \tag{35}$$

## A.8   MORE EXPERIMENTAL DETAILS OF NODE FEATURES' VISUALIZATION

To provide a quantitative analysis of the clustering visualizations corresponding to Figure 2, as discussed in the main paper, we utilize the Silhouette Coefficient. The specific silhouette scores for these visualizations are presented in the Table 5. The Silhouette Coefficient $s(i)$ for a single sample $i$ is defined as:

$$s(i) = \frac{b(i) - a(i)}{\max\{a(i), b(i)\}}.$$  (36)

$a(i)$ represents the average distance of sample $i$ to all other data points within the same cluster. This measures how well sample $i$ is assigned to its cluster (a smaller value indicates a better assignment). It is calculated as:

$$a(i) = \frac{1}{|C_I| - 1} \sum_{j \in C_I, i \neq j} d(i, j).$$  (37)

In this formula, $C_I$ is the cluster to which sample $i$ belongs, $|C_I|$ is the number of samples in cluster $C_I$ (the cardinality of $C_I$), and $d(i, j)$ is the distance (e.g., Euclidean distance) between sample $i$ and sample $j$ in the same cluster. If cluster $C_I$ has only one sample, $a(i)$ is typically considered to be 0 or undefined, though in practice, for a meaningful silhouette score, clusters should have more than one member.

$b(i)$ represents the smallest average distance of sample $i$ to all samples in any other cluster of which $i$ is not a member. This value quantifies the dissimilarity of sample $i$ to its "neighboring" closest cluster. It is calculated as:

$$b(i) = \min_{j \neq I} \left\{ \frac{1}{|C_j|} \sum_{k \in C_j} d(i, k) \right\}.$$

Here, the minimum is taken over all clusters $C_j$ where $i \notin C_j$. For each such cluster $C_j$, the average distance from $i$ to all points $k$ in $C_j$ is computed.

The overall Silhouette Coefficient for a dataset is the mean of $s(i)$ for all samples in the dataset.Ranging from -1 to 1, the Silhouette Coefficient indicates how well a data point fits its assigned cluster, where high values mean good fit, values near zero suggest it's on a cluster boundary, and negative values imply it might be misclustered. To assess the quality of learned node represen-

Table 5: Quantitative analysis of the node representation's visualizations.

| Method | Visualization Graph | Silhouette Coefficient |
|---|---|---|
| HOGRL | Figure 2a | -0.022 |
| HOGRL+GradConf | Figure 2f | 0.231 |
| BWGNN | Figure 2b | 0.019 |
| BWGNN+GradConf | Figure 2g | 0.164 |
| GTAN | Figure 2c | 0.050 |
| GTAN+GradConf | Figure 2h | 0.157 |
| RGTAN | Figure 2d | 0.054 |
| RGTAN+GradConf | Figure 2i | 0.166 |
| H2-FDetector | Figure 2e | 0.030 |
| H2-FDetector+GradConf | Figure 2j | 0.186 |

tations, we combine qualitative t-SNE visualization analysis Figure 2 with quantitative Silhouette Coefficient analysis Tabel 5. The t-SNE visualizations reveal that in one-shot scenarios, baseline models learn severely mixed and difficult-to-distinguish features for anomalous (orange points) and normal (cyan points) nodes. In contrast, GradConf significantly enhances feature discriminability. As a result, anomalous samples (orange points) form compact clusters that are clearly separated from normal ones (cyan points); samples within the same class are more tightly grouped while different classes are distinctly segregated, leading to clearer decision boundaries. The Silhouette Coefficients presented in Table 5 quantitatively support this: GradConf-enhanced models consistently achieve higher scores (e.g., HOGRL improves from -0.022 to 0.231; BWGNN from 0.019 to 0.164), indicating better-defined, more cohesive, and well-separated clusters. Collectively, these qualitative

and quantitative analyses demonstrate that GradConf effectively improves node representation quality under extremely one-shot conditions, thereby significantly enhancing the model's accuracy in distinguishing between anomalous and normal samples.

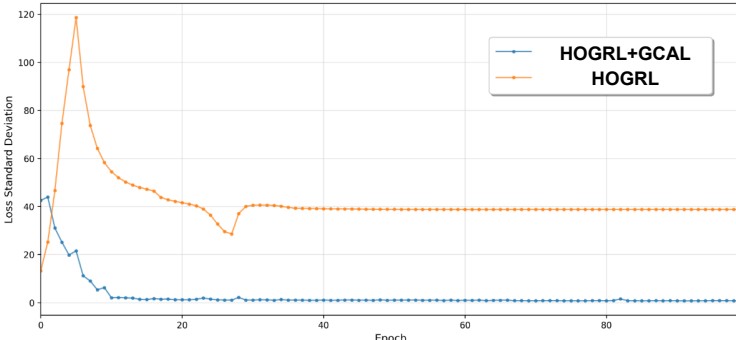

Figure 3: The training set loss standard deviation over epochs for the HOGRL model with and without the application of GCAL

### A.9 MORE EXPERIMENTAL DETAILS OF GCAL

This Figure 3 illustrates the Gradient-Confidence Aware Loss GCAL, and its direct effect on model training stability, using training loss standard deviation per epoch as the metric. The baseline HOGRL model without GCAL, the orange curve, displays high and fluctuating training loss standard deviation, signifying an unstable learning process. Conversely, the HOGRL model implementing GCAL, the blue curve, demonstrates a substantially lower and more consistent training loss standard deviation from the beginning, rapidly settling to a minimal, stable level.

This enhanced training stability stems directly from GCAL's core mechanism of adaptive sample weighting. By considering gradient information and prediction confidence, GCAL utilizes strategies like Focal weighting, dynamic class balancing, and gradient-based adjustments to effectively handle noisy pseudo-labels and data imbalances, this results in more reliable learning signals, a smoother optimization process, and significantly reduced variance in training loss, enabling more effective and consistent learning.

### A.10 MORE EXPERIMENTAL DETAILS OF PSEUDO-LABELS' QUALITY

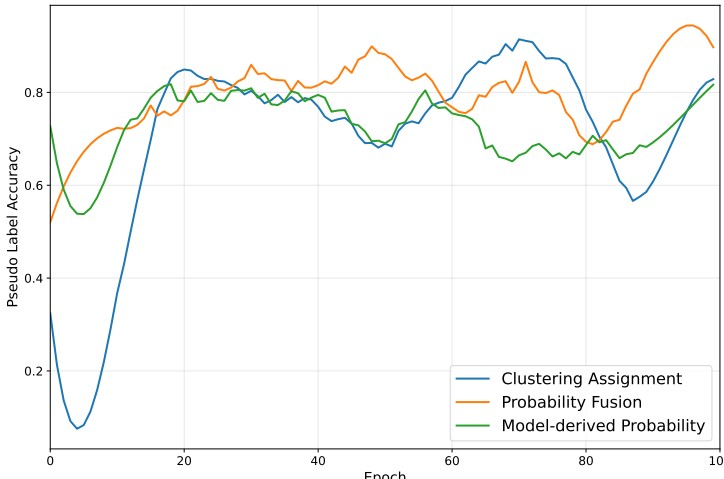

Figure 4: Comparison of pseudo-label accuracy across three different pseudo-label generation strategies

To optimize pseudo-label generation in few-shot learning scenarios, we compared three strategies: directly using raw model-derived probability, directly using clustering assignment, and a proposed fusion of both, with their dynamic pseudo-label accuracy during training illustrated in Figure 5. The analysis indicates that relying solely on model-derived probability offers training stability but a lower accuracy ceiling for pseudo-labels. Conversely, using only clustering results shows potential for higher accuracy but suffers from significant instability and fluctuations. The fusion strategy proposed in this GradConf effectively combines the advantages of the former two approaches. It mitigates the volatility of clustering by leveraging the stability of model outputs, while also harnessing the high-quality information from clustering to elevate the upper limit of pseudo-label accuracy. Consequently, the fusion strategy demonstrated the best overall performance throughout the training process, achieving smoother, more stable, and ultimately the highest pseudo-label accuracy, thereby providing more high-quality pseudo-labels for the model.

## A.11 WHY USE KL DIVERGENCE IN PCSC ?

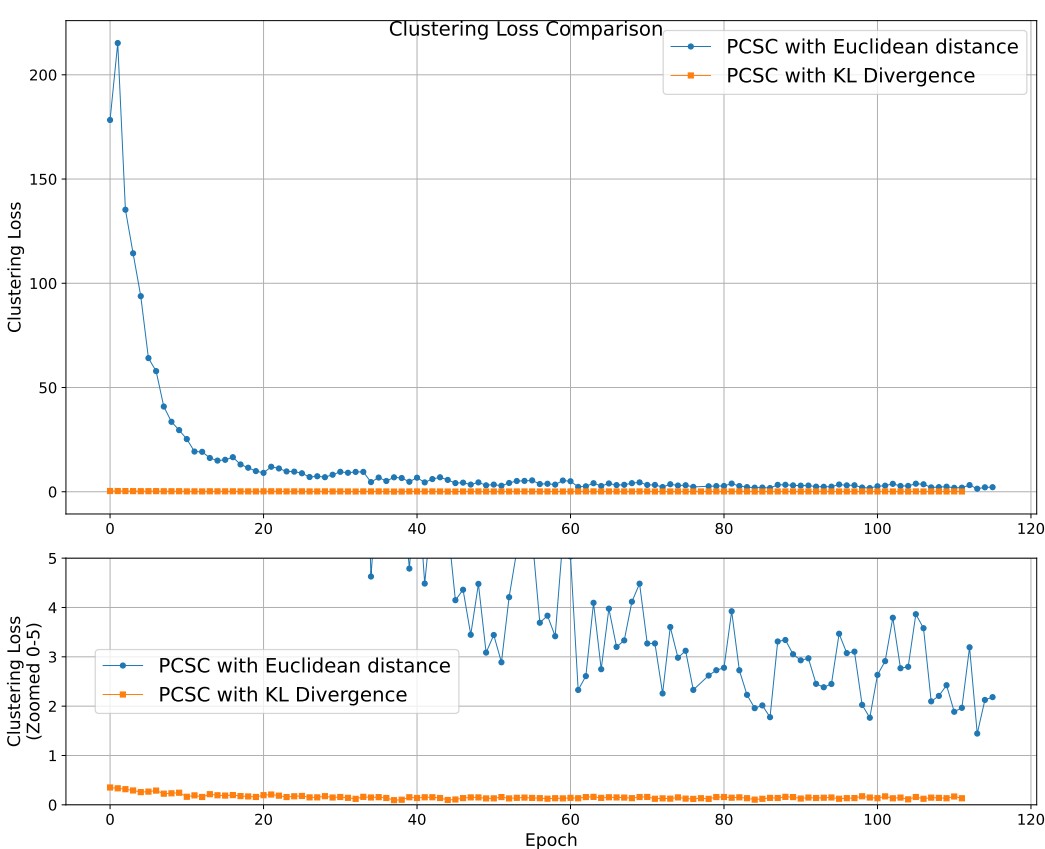

Figure 5: Comparison of Convergence and Stability for PCSC Clustering Loss: Euclidean Distance vs. KL Divergence

The measure used in PCSC's clustering loss critically impacts training stability and pseudo-label quality. As shown in Figure 5, a Euclidean distance-based loss, despite an initial sharp drop from a very high starting value (around 220), exhibited persistent and significant volatility throughout training, still oscillating roughly between 1.0 and 5.0 even after many epochs. This erratic behavior suggests unstable gradients and inconsistent cluster formation. In stark contrast, the KL Divergence-based loss started much lower (around 0.5), converged rapidly within the first few epochs to a minimal (around 0.1 or less) and exceptionally stable value, maintaining negligible fluctuations. This clearly demonstrates that KL Divergence provides a superior learning signal, fostering coherent cluster development and thus enabling the generation of higher-quality, more reliable pseudo-labels, which are essential for the PCSC module's overall effectiveness.

### A.12 SEMI-SUPERVISED EXPERIMENTS

The experimental setup is consistent with that described in our main paper and appendix, with the only modification being that we changed the one-shot setting to a semi-supervised one where we retain 10% of the labeled samples in the training set. The results are as follows: An asterisk (*) indicatesstatistical significance (with p<0.05) when comparing GradConf to the best baseline results.

Table 6: Semi-supervised experiments

| Model | Amazon | | | | | YelpChi | | | | | S-FFSD | | | | |
|---|---|---|---|---|---|---|---|---|---|---|---|---|---|---|---|
| | AUC | F1 | Gmean | ACC-0 | ACC-1 | AUC | F1 | Gmean | ACC-0 | ACC-1 | AUC | F1 | Gmean | ACC-0 | ACC-1 |
| H2-FDetector | 86.03* | 55.98* | 75.74* | 64.85* | 88.45* | 72.68* | 53.80* | 67.60* | 67.73* | 67.48* | 70.22* | 49.96* | 65.83* | 60.15* | 72.04* |
| RGTAN | 89.94* | 58.90* | 78.01* | 69.31* | 87.80* | 76.70* | 49.34* | 67.37* | 56.99* | 79.64* | 72.20* | 31.03* | 48.68* | 27.17* | 87.20* |
| GTAN | 91.00* | 83.34* | 82.58* | 96.71* | 70.52* | 77.82* | 53.48* | 71.05* | 63.68* | 79.27* | 73.31* | 45.56* | 64.18* | 50.56* | 84.16* |
| BWGNN | 92.26* | 55.04* | 75.88* | 62.53* | 92.07* | 79.01* | 58.85* | 72.58* | 74.06* | 71.13* | 63.62* | 45.44* | 60.60* | 53.77* | 68.29* |
| HOGRL | 83.49* | 86.14* | 85.37* | 98.79* | 73.78* | 77.44* | 42.57* | 62.41* | 44.05* | 88.41* | 61.48* | 28.89* | 46.16* | 23.92* | 89.06* |
| SpaceGNN | 87.85* | 84.34* | 79.64* | 94.40* | 67.08* | 60.65* | 52.17* | 39.07* | 86.70* | 16.18* | 60.48* | 56.31* | 39.83* | 15.54* | 92.87* |
| H2-FDetector+GradConf | 95.41* | 74.63* | 87.98* | 85.49* | 90.54* | 82.83* | 58.18* | 74.95* | 70.55* | 79.65* | 74.05* | 54.42* | 69.80* | 67.14* | 72.56* |
| RGTAN+GradConf | 95.48* | 79.39* | 89.38* | 91.24* | 87.56* | 85.20* | 64.00* | 78.18* | 78.86* | 77.51* | 74.84* | 51.00* | 67.81* | 60.76* | 75.68* |
| GTAN+GradConf | 96.56* | 84.05* | 91.54* | 94.28* | 88.87* | 86.53* | 64.87* | 80.12* | 77.75* | 82.32* | 75.28* | 42.81* | 61.72* | 45.41* | 83.89* |
| BWGNN+GradConf | 97.72* | 89.43* | 91.17* | 89.76* | 92.60* | 86.72* | 63.84* | 79.16* | 77.81* | 80.55* | 70.75* | 42.86* | 61.99* | 45.33* | 84.76* |
| HOGRL+GradConf | 98.83* | 90.30* | 94.13* | 89.00* | 99.56* | 87.38* | 60.89* | 77.32* | 73.94* | 80.85* | 77.44* | 42.57* | 62.41* | 44.05* | 88.41* |
| SpaceGNN+GradConf | 95.50* | 89.10* | 90.80* | 93.20* | 88.50* | 78.50* | 61.50* | 70.20* | 75.80* | 68.90* | 71.80* | 60.15* | 65.50* | 55.40* | 80.10* |

Our semi-supervised experiments show that adding GradConf to the baseline models leads to significant performance improvements and a more balanced predictive capability for positive and negative samples. This performance even surpasses the level of the fully-supervised baseline models. These results validate our model's effectiveness and scalability beyond the extreme one-shot scenario and confirm its practical applicability and generalization performance in more realistic fraud detection environments.

## B PARAMETER SENSITIVITY ANALYSIS

To comprehensively evaluate the stability of the GradConf framework and the specific impact of its key hyperparameters on performance, we conducted a series of detailed parameter sensitivity experiments.

**Gradient-Confidence Aware Loss (GCAL) Parameters**: The GCAL module is designed to adap-

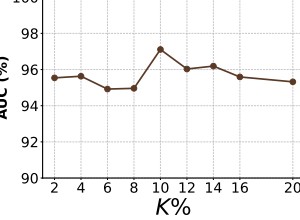

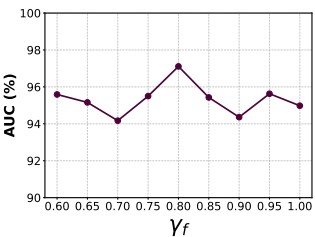

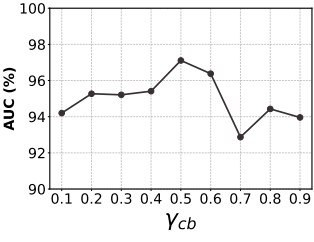

(a) Hardest Samples Parameter $K\%$

(b) Focal Weighting Parameter $\gamma_f$

(c) Class Balancing Parameter $\gamma_{cb}$

Figure 6: Hyper-parameters in GCAL.

tively balance the contributions of positive and negative sample pairs by coupling global gradient signals with instance-level confidence, thereby mitigating optimization bias caused by extreme class imbalance. The sensitivity of its key hyperparameters is depicted in Figure 6 of the Appendices.

Regarding the Hardest Samples Parameter K%, which is utilized in the dynamic class balancing weight calculation to determine the proportion of hardest samples for estimating class weights, Figure 1a of the Appendices shows that the AUC performance of GradConf remains relatively stable when K% varies between 2% and 20%. Performance peaks around K%=10%, which is the default setting in our paper. A K% that is too small might not sufficiently capture the dynamic changes in class difficulty, while a K% that is too large could introduce noise, impacting weight estimation accuracy. When K is set to 10%, the results in Figure 6a indicates best performance on AUC.

For the Focal Weighting Parameter $\gamma_f$, it is designed to modulate focus on hard versus easy samples, a ratio within the 0-2.5 range. A $\gamma_f > 0$ allows more emphasis on samples with lower confidence. If $\gamma_f$ is set too low, the model may not sufficiently prioritize challenging samples, potentially hindering its ability to learn subtle anomaly patterns. Conversely, if $\gamma_f$ is set too high, the model might over-focus on a few extremely difficult or noisy samples, which can lead to suboptimal learning of the overall data distribution and a slight performance dip. The results in Figure 6b indicate that a value of 0.80 (the default of 4/5) is most suitable for $\gamma_f$, providing an optimal balance in emphasizing difficult samples.

The Class Balancing Parameter $\gamma_{cb}$ controls the intensity of the class weight adjustment to alleviate class imbalance. Figure 6c indicates high AUC when $\gamma_{cb}$ is between 0.3 and 0.7, with optimal performance around 0.5 (the default value of 1/2). If $\gamma_{cb}$ is too low, class balancing is diminished; if too high, it might excessively amplify the minority class influence. The results confirm good adaptability around the recommended value (set to 1/2).

**Pseudo-label Clustering Self-Correction (PCSC) Parameters**: The PCSC module iteratively re-

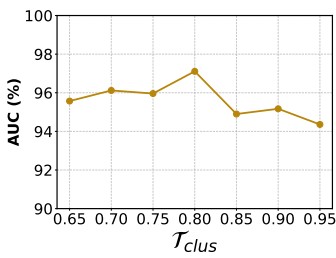
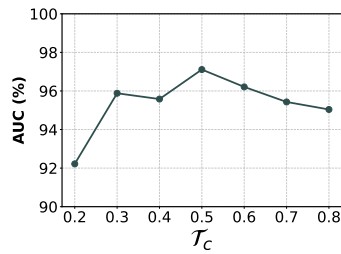

(a) Parameter $\tau_{clus}$ for the sharpness of the soft assignments

(b) Parameter $\tau_c$ for the sensitivity to dissimilarity between centroids

Figure 7: Hyper-parameters in PCSC.

fines pseudo-label quality through learnable cluster centroids and a structure-aware self-revision mechanism, addressing pseudo-label instability in one-shot scenarios. The impact of its key hyper-parameters is shown in Figure 7 of the Appendices.

The Sharpness Parameter $\tau_{clus}$ appears in the Gumbel-Softmax calculation for soft cluster assignments $q_{u,j}$ and controls the sharpness of these assignments. Observing Figure 7a of the Appendices, the AUC performance is stable and high when $\tau_{clus}$ varies between 0.75 and 0.90, with optimal results around 0.80 (the default value of 4/5). A smaller $\tau_{clus}$ leads to smoother assignments, while a larger $\tau_{clus}$ makes assignments closer to hard assignments. The experiments suggest that moderate sharpness (set to 4/5) beneficially balances assignment determinism and flexibility.

The Centroid Dissimilarity Sensitivity $\tau_c$ is a temperature hyperparameter in the inter-cluster loss $\mathcal{L}_{inter}$ controlling sensitivity to dissimilarity between centroids. Figure 7b shows high AUC for $\tau_c$ between 0.3 and 0.6, peaking around 0.5 (the default value of 1/2). A smaller $\tau_c$ encourages greater separation between cluster centroids, whereas a larger value tolerates less separation. Excessively high or low values could lead to suboptimal cluster structures, affecting pseudo-label quality. v
**Logits Adversarial Perturbation (LAP) Parameters**: The LAP module enhances sensitivity to anomaly instances and improves generalization under distributional sparsity by introducing adversarial perturbations to logits. The sensitivity of its core hyperparameters is detailed in Figure 8.

For the Base Single-step Strength $\alpha_0$, which serves as the base setting for single-step perturbation strength in LAP (dynamically adjusted based on class imbalance and sample hardness), Figure 8a shows high AUC performance when $\alpha_0$ is within [0.02, 0.05]. It peaks at $\alpha_0 = 0.03$ (the default value). An $\alpha_0$ that is too small may result in insufficient perturbation, while an overly large $\alpha_0$ could introduce excessive noise.

Regarding the Base Total Steps $S_{0\_iter}$, defining the foundational number of iteration steps for perturbing logits in LAP, Figure 8b illustrates stable and superior AUC performance when $S_{0\_iter}$ ranges between 15 and 30 steps. Optimal performance is observed around 20 steps (the default value). Too few steps might not adequately enhance robustness, whereas an excessive number could increase computational overhead and noise.

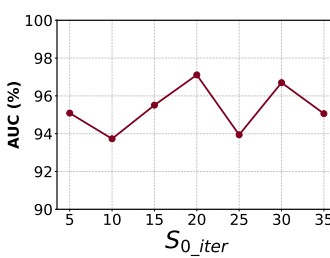
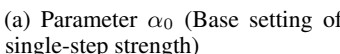
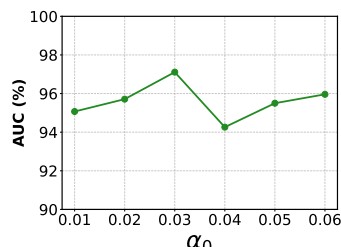

(a) Parameter $\alpha_0$ (Base setting of single-step strength)

(b) Parameter $S_{0\_iter}$ (Base setting of total steps)

Figure 8: Hyper-parameters in LAP.

**Model Optimization Balance Parameters:** The parameter $\mu$ (Laine & Aila, 2016) critically bal-

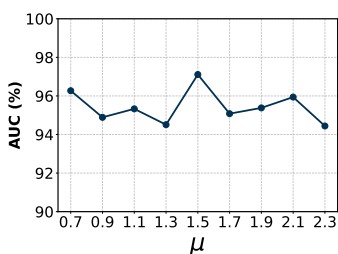

Figure 9: Model Optimization Balance Parameters $\mu$.

ances components in GradConf's total objective function, $\mathcal{L}_{total} = \mathcal{L}_{sup} + \mathcal{L}_{cls} + \mu(\mathcal{L}_{cons} + \mathcal{L}_{gcal} + \mathcal{L}_{lap} + \mathcal{L}_{pcsc})$. In our framework, this balance parameter $\mu$ is dynamically calculated for each epoch, denoted as $\mu_{epoch}$, using the following formula:

$$\mu_{epoch} = \mu_{base} \cdot \exp\left(-5.0 \cdot \left(1.0 - \frac{clip(epoch, 0, max\_epoch)}{max\_epoch}\right)^2\right) \tag{38}$$

Here, $\mu_{base}$ is the base hyperparameter that we tune, and it corresponds to the $\mu$ discussed in this section. The $clip(epoch, 0, max\_epoch)$ function ensures that the current epoch stays within the bounds of 0 and $max\_epoch$. This means that $\mu_{epoch}$ starts at $\mu_{base}$ and its value is adjusted through-out the training process.In our specific setup, $max\_epoch$ within Eq.(1) is set to 500 (even if actual training epochs are set to 100) for a more gradual $\mu_{epoch}$ ramp-up towards $\mu_{base}$. This approach, inspired by previous work (Laine & Aila, 2016) highlighting the importance of a slow ramp-up for unsupervised components, prioritizes initial learning from supervised signals and reduces early noise impact from unlabeled data or auxiliary tasks. Figure 4 shows the impact on AUC as this base $\mu$ varies. A low $\mu_{base}$ diminishes the effectiveness of GCAL's adaptive re-weighting for class imbalance, PCSC's pseudo-label refinement for leveraging unlabeled data, and LAP's adversarial robustness enhancement for anomaly sensitivity. This can hinder the model's adaptation to complex anomaly features and effective use of sparse supervision. Conversely, a high $\mu_{base}$ risks overshad-owing fundamental supervised signals from the minimal true labels, potentially causing instability in the challenging one-shot learning context. Empirical results indicate that $\mu_{base}$ (set to 1.5 in our method) provides the optimal balance, ensuring all GradConf components effectively contribute to robust one-shot anomaly detection.

In summary, the sensitivity analyses for the GCAL, PCSC, and LAP components, along with the model optimization balance parameter $\mu$, demonstrate that GradConf is generally robust. Variations in our key hyperparameters, when kept within reasonable ranges around their optimal settings, still allow the model's performance to remain at a commendably high level overall. The primary goal of this tuning process is to identify the specific value that empowers GradConf to achieve its best possible one-shot anomaly detection performance under the demanding conditions of extremely limited supervision.

## C MORE EXPERIMENTS DETAILS ABOUT EFFICIENCY

All experiments were conducted on a single H20 GPU. Our framework is designed as a plug-and-play module that can be integrated with various baseline models. During training, the integration of our method incurs overhead due to its dual-branch architecture, increasing the training duration by approximately 1.5 times and doubling the GPU memory usage compared to the baseline. Conversely, for inference, our framework is highly efficient and does not introduce any additional computational or memory overhead; thus, the inference time and VRAM consumption are identical to those of the baseline model.

## D USE OF LLMS

In the development of this paper, Large Language Models (LLMs) were utilized solely for aiding and polishing the writing of the manuscript. Specifically, LLMs were employed to refine the clarity and accuracy of English expressions across sections (e.g., technical descriptions in the methodology) and to check for grammatical errors. Details regarding the scope of LLM use are consistent with ICLRs transparency guidelines and are briefly summarized herein.

