# OpenReview forum: "One Anomaly to Catch Them All: Graph-Based One-Shot Anomaly Detection via GradConf"
_ICLR.cc/2026/Conference — ICLR 2026 Conference Withdrawn Submission_

### Official Review · Reviewer_geEu · 2025-10-26

**Soundness:** 2
**Presentation:** 2
**Contribution:** 2
**Rating:** 2
**Confidence:** 4

**Summary:**

This paper focuses on graph anomaly detection under extremely low supervision and proposes a method named GradConf. In this setting, only one normal node and one abnormal node are provided. To address this problem, this paper utilizes pseudo-labeling to enlarge the supervision. Specifically, three modules are designed which are Gradient-Confidence Aware Loss, Designing a Pseudo-label Clustering
Self-Correction module and Logits Adversarial Perturbation.

**Strengths:**

1. The studied problem is important and interesting.

2. Three modules are designed specifically for addressing GAD with extremely low supervision.

3. The proposed method can be applied to other baselines and improve their performance under the proposed setting.

**Weaknesses:**

1. The paper is not well-written. For each component, it is hard to understand the details and the role of each operation.

2. Some symbols are used incorrectly, resulting in confusion for readers.

3. The authors propose to construct two augmentations for the original graph. Are they important for the proposed three modules?

4. The authors argue that the proposed framework constructs a graph structure where nodes represent entities and edges denote associative relationships. However, the studied graph already consists of nodes and edges.

**Questions:**

Please see the weaknesses.

---

### Official Review · Reviewer_WahV · 2025-10-27

**Soundness:** 2
**Presentation:** 1
**Contribution:** 2
**Rating:** 2
**Confidence:** 4

**Summary:**

This paper studies the one-shot graph anomaly detection issue. The core challenges of the issue are 1) extremely low labelling rates, 2) weak graph homophily, and 3) extreme class imbalance. To address these challenges, this paper proposes the gradient-aware loss, clustering-based pseudo-label correction and adversarial perturbation. Experiments on 6 datasets show the proposed method GradConf achieves good performance.

**Strengths:**

1.	This paper focuses on graph anomaly detection with only one pair of normal and abnormal nodes, which is a challenging problem.
2.	Three modules with specific purposes are designed for class imbalance and pseudo labelling.
3.	By applying the proposed method into baselines, the performance of these baselines is improved.

**Weaknesses:**

1.Since the number of anomalies is significantly smaller than the normal nodes, it will be difficult and time-consuming to collect abnormal nodes in practice, limiting the claim of the paper.
2.The proposed method is very complicated and hard to understand the details of each component, especially the logits adversarial perturbation. In addition, the authors should use different symbols for different objects. For example, the node and graph augmentation are both denoted as t.
3. More analysis should be provided on the utilization of two augmentation views for this task.
4. Although you have proposed several different loss functions to address distinct issues in your method, the motivation behind each loss is not clearly articulated. It remains unclear why each loss component is suitable for tackling its corresponding challenge. As a result, the method appears to be a simple combination of multiple losses rather than a principled integration driven by specific design rationales.
5. Although you provide a theoretical analysis, there is a substantial gap between the theory and the proposed method. In particular, some key terms appearing in your loss functions are not grounded in the theoretical framework, leaving their necessity and interpretation unclear. Moreover, the mathematical proof of your core theoretical result is missing, which raises concerns about the correctness and completeness of your theoretical claims. Therefore, the current theoretical section is too weak to convincingly support your method.

**Questions:**

Please see the weaknesses, and
1.	Please provide clear explanations on different term of your loss to show its principle why this term can address your different challenge. If necessary, please give clear experimental observations.
2.	Please rebuild your theoretical part to ensure your theory is not a fake theory for one-shot GAD.

---

### Official Review · Reviewer_2Qms · 2025-10-30

**Soundness:** 3
**Presentation:** 2
**Contribution:** 2
**Rating:** 2
**Confidence:** 4

**Summary:**

This paper proposes the GradConf framework for one-shot graph anomaly detection, improving detection performance through three modules: Gradient-Confidence Aware Loss (GCAL), Pseudo-label Clustering Self-Correction (PCSC), and Logits Adversarial Perturbation (LAP). Experiments on 6 datasets validate the method's effectiveness, achieving or exceeding fully supervised performance using only one pair of labeled samples. Overall, the paper addresses the specific problem of graph anomaly detection under extremely limited supervision through the integration and refinement of multiple existing modules.

**Strengths:**

1. The problem of difficult sample annotation in graph anomaly detection scenarios addressed by the authors has practical significance.
2. The three modules designed to address three key challenges demonstrate reasonable design rationale, relatively clear method descriptions, and complete theoretical foundations.
3. The experimental details are clearly and comprehensively described, with excellent experimental results and impressive performance improvements.

**Weaknesses:**

1. The proposed method includes three independent modules and numerous hyperparameters, but ablation experiments are only conducted at the module level, lacking more detailed and systematic evaluation of individual components within each module.
2. The method has high complexity. The paper only mentions increased training time and memory after integrating the modules, but lacks more detailed complexity analysis.
3. The authors claim that the proposed method even exceeds fully supervised methods on some datasets, but lack analysis and explanation of this phenomenon. Does this indicate overfitting issues in fully supervised methods, or does the proposed method have other advantages?
4. The authors need to further clarify technical details in the methodology section, see Questions 1-3 below.
5. The authors claim that the framework "enhances the model's generalization ability to detect unknown heterogeneous anomalies," but experiments on standard benchmarks do not seem to sufficiently validate this generalizability.

**Questions:**

1. The description of the PCSC module is unclear. In Section 3.4, the initialization of clustering centers depends on $H_{orig}$, but there is no explanation of how H_orig is obtained or updated.
2. In Equation (9), the Gumbel-Softmax with $\tau$ closer to 1 introduces higher randomness. The authors set it to 4/5, so how is the stability of sampled pseudo-labels guaranteed?
3. Pseudo-label generation depends on clustering results, while clustering optimization depends on pseudo-labels. How is the convergence of this circular dependency guaranteed?

---

### Official Review · Reviewer_KjNW · 2025-10-30

**Soundness:** 2
**Presentation:** 2
**Contribution:** 2
**Rating:** 2
**Confidence:** 5

**Summary:**

This paper proposes an extremely low-supervision setting in which only a single anomalous sample is available for anomaly detection. It introduces GradConf, which is jointly optimized using a gradient confidence–aware loss designed to dynamically balance positive and negative samples. Furthermore, a clustering-based self-correction module is developed to refine the pseudo labels. Finally, a logit adversarial perturbation strategy is employed to enhance the model’s sensitivity by injecting controlled perturbations.

**Strengths:**

(1) This paper is easy to understand and introduces a more challenging setting for semi-supervised GAD.

(2) It primarily serves to demonstrate that the proposed GradConf performs best under this particular condition.

**Weaknesses:**

(1) This setting is interesting for research but not very practical. In real-world applications, however, scenarios with only a single available anomaly are quite uncommon. Although the authors focus on the one-shot setting, conducting experiments under 2-shot and 5-shot conditions is necessary to evaluate the model’s generalization and stability. I suspect that the observed improvement does not primarily stem from the one-shot anomaly itself, but rather from the data augmentation and pseudo-labeling strategies, which could yield similar benefits even outside the one-shot setting.

(2) The methodology is not well motivated. There appears to be little connection between the clustering used for pseudo-label assignment and the augmentation consistency component.  There are no underlying relations in the gradient-aware loss, clustering-based pseudo-label correction, and adversarial perturbation to tackle class imbalance.  Moreover, the approach heavily depends on pseudo-labeling, which is not specifically designed for this setting, thereby limiting its novelty and contribution. Essentially, the method still treats the task as a conventional imbalanced classification problem, making the method seem like a specialized data augmentation.

(3) If the selected anomaly sample varies, will the model’s performance fluctuate considerably, as the quality of the known anomaly can strongly influence the overall results? Moreover, the comparison with few-shot GAD approaches such as MetaGAD [Ref1] and Meta-GDN [Ref2] is lacking; restricting the evaluation to fully supervised baselines is inadequate for a thorough assessment. It is also worth noting that investigating the few-shot setting is fundamentally different from addressing an augmentation-based imbalanced classification problem.

[Ref1]  "MetaGAD: Meta Representation Adaptation for Few-Shot Graph Anomaly Detection." 2024 IEEE 11th International Conference on Data Science and Advanced Analytics (DSAA). IEEE, 2024.
[Ref2] "Few-shot network anomaly detection via cross-network meta-learning." Proceedings of the Web Conference 2021.

(4) Six loss functions are jointly optimized, which may make it challenging for them to effectively coordinate with one another. The experimental section in the main paper is rather brief, and some of the analyses lack sufficient detail. The authors are encouraged to reorganize and expand the overall presentation of the paper to improve clarity and completeness.

**Questions:**

See above **Weakness**

---

### Note · Authors · 2025-11-30

I have read and agree with the venue's withdrawal policy on behalf of myself and my co-authors.